# Zero-shot World Models via Search in Memory

**Federico Malato**
School of Computing
University of Eastern Finland
Joensuu, FI 80101
`federico.malato@uef.fi`

**Ville Hautamäki**
School of Computing
University of Eastern Finland
Joensuu, FI 80101
`ville.hautamaki@uef.fi`

## Abstract

World Models have vastly permeated the field of Reinforcement Learning. Their ability to model the transition dynamics of an environment have greatly improved sample efficiency in online RL. Among them, the most notorious example is Dreamer, a model that learns to act in a diverse set of image-based environments. In this paper, we leverage similarity search and stochastic representations to approximate a world model without a training procedure. We establish a comparison with PlaNet, a well-established world model of the Dreamer family. We evaluate the models on the quality of latent reconstruction and on the perceived similarity of the reconstructed image, on both next-step and long horizon dynamics prediction. The results of our study demonstrate that a search-based world model is comparable to a training based one in both cases. Notably, our model show stronger performance in long-horizon prediction with respect to the baseline on a range of visually different environments.

## 1 Introduction

World Models (WMs) [7] have played a fundamental role in the most recent advances in reinforcement learning (RL) [19]: their ability to predict future states of a process, along with the possibility to enhance planning [3] have led to tremendous achievements in autonomous agents. WMs have demonstrated their impressive capabilities in a range of tasks, from playing small scope [9] and open ended [10] video games, to autonomous driving [4], up to robotic control [20]. Recently, WMs have shown also to be adaptable to diverse domain [10], enlarging the pool of their potential application even further. Moreover, recent discussions hint a possible connection between world models and Large Language Models (LLMs) [3].

However, learning the dynamics of an environment is not a trivial task: modeling temporal dynamics implies learning an underlying causal structure of the environment [5, 16], which in turn implies strong generalization and adaptability skills. As such, WMs typically require massive amounts of data to construct a solid internal model. Moreover, they are subject to hallucinations and error accumulation in long-term prediction [3], which limit the effectiveness of the prediction. To ease this limitation, more complex architectures are required [5, 3], which in turn require more data and more computational resources.

In this paper, we introduce an alternative formulation of World Models derived from similarity search and probabilistic modeling, which we refer to as *zero-shot World Models* due to their independency from a standard training procedure. The contributions our study are three-fold: first, we explore the theoretical feasibility of a memory-based world model; second, we explore the capabilities and compare it to a well-known, learning-based model; third, we determine a range of tasks for which such models are applicable, and clearly state situations where they are unfeasible. We remark that our aim is to explore a valid alternative to current world models, focusing on the task of dynamics prediction and reconstruction, rather than action selection.

39th Conference on Neural Information Processing Systems (NeurIPS 2025).

## 2 Related Work

Our study draws its main inspirations from PlaNet [8] and its evolution Dreamer [9, 10]. Moreover, we base our study on previous work on similarity search, namely [13, 15, 12, 1]. In this Section, we briefly revise and introduce the main concepts of each work.

In [8], authors propose PlaNet, a model-based RL agents that learns to plan from pixels. In their study, authors model the state of an environment as composed by a deterministic and a stochastic part. To successfully predict future states, they define a recurrent *state-space model* (SSM) composed of a variational autoencoder (VAE) [11], an observation model $\mathbb{P}(o_t|s_t)$, a transition model $\mathbb{P}(s_t|s_{t-1}, a_{t-1})$, a reward model $\mathbb{P}(r_t|s_t)$ and a recurrent, deterministic state model $h_t = f(h_{t-1}, s_{t-1}, a_{t-1})$. While leaving the general structure of the model substantially unaltered, evolutions of PlaNet introduce, respectively, a discrete underlying distribution of the latent space [9] and an improved loss for more stable predictions [10].

In [1], authors define *locally weighted learning* (LWL), a framework to train a model for continuous control by using an ensemble of local models. In particular, the dataset is projected into a metric state space and divided into neighborhoods, hence producing subsets of closely related data. Then, a local model is trained on each specialized dataset. Finally, an agent selects actions by querying each local model and performing a weighted average over their answers.

In [15], authors apply LWL in the context of robotic manipulation. In particular, they demonstrate that separating representation and behavior learning improves robustness in robots. In their study, they pre-train an encoder on a dataset of images to extrapolate a suitable latent dataset for their task; then, they apply LWL to predict actions from an ensemble of local agents.

Zero-shot Imitation Policy (ZIP) [13] illustrates how an agent can be successfully controlled without learning even in open-ended tasks: using tasks from Minecraft [6] as their benchmark, they encode temporally extended latents in latent space using Video PreTraining (VPT) [2] and track the divergence of the current state from a retrieved reference state in latent space. For each timestep, they copy the action of the retrieved sequence. When the two states become too distant, a new search is repeated, and a new sequence of actions is followed.

In [12], authors combine ideas from LWL and ZIP to adapt a learning-based agent online, leveraging only Bayesian statistics. In detail, authors pair a pre-trained imitation learning agent with a search-based policy. At each timestep, the search-based policy retrieves a batch of latents similar to the current state, from which they build a probability distribution for the current state following the "suggestions" of an expert. Then, authors combine the imitation policy and the expert policy suggestions by building a posterior distribution over actions from the two.

## 3 Zero-shot World Models

We derive our approach by combining previous studies on similarity search [13, 15, 12, 1] with probabilistic modeling, specifically Variational Autoencoders (VAEs) [11]. We state our problem as follows: given a dataset $\mathcal{D}$ of state-action pairs $(x_t, a_t) \in \mathcal{D}$, where $x_t$ is an RGB image representing the state of a system at timestep $t \in [0, T]$, can we predict the transition dynamics $\mathbb{P}(x_{t+1}|x_t, a_t)$ of an environment *without* learning them?

Initially, we train a VAE to reconstruct images $x_t$ from our dataset $(x_t, a_t) \sim \mathcal{D}$. Importantly, VAEs operate in two consecutive steps: first, given an image $x_t$, a stochastic, latent representation $\hat{z}_t \sim q_\phi(\boldsymbol{Z_t}|x_t)$ is obtained. Then, an approximation of the initial image $\hat{x} \sim p_\theta(\boldsymbol{X_t}|z_t)$ is recovered by passing $z$ through the decoder. After training, we encode each $x_t$ to obtain a latent dataset $\mathcal{Z} = \{(z_t, a_t)|z_t \sim q_\phi(\boldsymbol{Z_t}|x_t), (x_t, a_t) \sim \mathcal{D}\}$. Similar to previous work [11], we choose $q_\phi(\boldsymbol{Z_t}|x_t)$ to be a multivariate Normal distribution $\mathcal{N}(\mu_\phi(x_t), \Sigma_\phi(x_t))$

Following previous work in similarity search [13, 15], given a latent $z_t$ at timestep $t$, we can easily produce a batch of $K$ relevant samples $\{(\tilde{z}_{1,\tau}, a_{k,\tau})\}_{k=1}^K, \tau \in [0, T]$. Moreover, assuming that the retrieved latents are sufficiently close to each other, we could estimate $q_\phi(\boldsymbol{Z_t}|x_t)$. From this, a natural question arises: what if, instead of approximating $q_\phi(\boldsymbol{Z_t}|x_t)$ directly, we moved one step forward in time from each of our retrieved latents? Namely, what would happen if we tried to approximate $q_\phi(\boldsymbol{Z_{t+1}}|x_{t+1})$ from $\{(\tilde{z}_{k,\tau+1}, a_{k,\tau+1})\}_{k=1}^K$?

Intuitively, if we could do that, then we could sample a general representation of the *expected next state in time*, $z_{t+1} \sim q_\phi(\boldsymbol{Z_{t+1}}|x_{t+1})$. We point out that since the pairs $(x_t, a_t)$ represent states and actions of a system, we could view them as trajectories $\mathcal{T} = \{(x_t, a_t, x_{t+1})\}_{t=0}^T$. By exploiting this simple trick, we can reshape also the latent dataset $\mathcal{Z}$ to store transitions $(z_t, a_t, z_{t+1})$ instead. Now, whenever we search for latents that are similar to $z_t$, we retrieve a batch of relevant transitions $\{(\tilde{z}_{k,\tau}, a_{k,\tau}, \tilde{z}_{k,\tau+1})\}_{k=1}^K$. In summary, given a reference latent $z_{\text{ref},t}$ and a latent dataset of transitions $\mathcal{Z}$, we *can* obtain a new latent $\hat{z}_{t+1}$ by leveraging similarity search and the stochastic representation of a VAE. That is, we can predict an approximation of the next state $\hat{z}_{t+1} \sim \mathbb{P}(z_{t+1}|z_t)$ from the current one.

Notably, the last statement is very similar to the transition dynamics $\mathbb{P}(s_{t+1}|s_t, a_t)$ of a Markov Decision Process (MDP) [19]. Hence, we advance our theory in this direction: can we approximate the transition dynamics by repeating the same procedure, but imposing a certain action? Following from our previous remarks, the answer to this question is yes: given a reference pair $(z_{\text{ref},t}, a_{\text{ref},t})$, we can either perform an unconstrained search on $z_{\text{ref},t}$ to obtain a batch $\{(\tilde{z}_{k,\tau}, a_{k,\tau}, \tilde{z}_{k,\tau+1})\}_{k=1}^K$ and subsequently remove latents for which $a_{k,\tau} \neq a_{\text{ref},t}$, or directly restrict the search to $\mathcal{Z}_{a_{\text{ref},t}} = \{(z_\tau, a_\tau, z_{\tau+1}) \in \mathcal{Z}|a_\tau = a_{\text{ref},t}\}$.

Until now, we have established that given a pair $(z_t, a_t)$ from a set of trajectories $\mathcal{Z}$, we can approximately predict how the state will evolve one step in the future by approximating $\mathbb{P}(z_{t+1}|z_t, a_t)$ in latent space through similarity search and probabilistic modeling. In summary, we have effectively recovered the functionality of a next-state predictor without learning. Still, an effective World Model would ideally extend its prediction to a longer horizon $h = t + \Delta t, \Delta t > 1$, as demonstrated by PlaNet, Dreamer, and similar models [3]. Therefore, how can we extend our idea to include long-horizon predictions?

Intuitively, we could apply the same "retrieve & reconstruct" procedure iteratively. More formally, given an initial pair $(z_t, a_t) \sim \mathcal{Z}$ and the next predicted state $\hat{z}_{t+1} \sim \mathbb{P}(z_{t+1}|z_t, a_t)$, we could search a new batch $\{(\tilde{z}_{k,\tau}, a_{k,\tau}, \tilde{z}_{k,\tau+1})\}_{k=1}^K$ using $\hat{z}_{t+1}$ and estimate $\mathbb{P}(z_{t+2}|z_{t+1}, a_{t+1})$, effectively evolving the state for a second step in the future. We provide some implementation details of this procedure in the supplementary material of this paper.

However, one problem arises: how do we select $a_{t+1}$? From our latest search, we have retrieved a batch of actions $\{a_{1,t+1}, \ldots, a_{k,t+1}\}$ which, in general, will not be all equal. One solution to this includes estimating a probability distribution over actions $\mathbb{P}(a_{t+1}|\hat{z}_{t+1})$ and sampling from it, similarly to [12] for the discrete case and [15] for continuous actions. Although this is a valid alternative, it implies moving the focus towards the problem of action selection, which is outside the scope of this paper. We solve the problem by making an additional assumption that highlights the effects of dynamics prediction. To avoid the fuzziness that comes from sampling future actions, we assume complete access to future actions. As such, given an initial latent $z_t$, we retrieve a sequence of *deterministic* actions $\{a_t, \ldots, a_{t+h}\}$ up to an horizon $h = t + \Delta t$. We remark that, while this assumption would make our approach unfeasible for an action selection process, we are interested in assessing the quality of dynamics prediction over time.

Intuitively, searching constitutes a major factor in the success of our approach. As such, without completeness, we propose three different structures for searching. In the following Subsections, we highlight the unique traits of each and discuss their strengths and weaknesses. A visualization of these modalities for retrieval, along with a visual representation of the different search procedures is provided in Figure 1.

### 3.1 Rollout buffer

In a MDP, a trajectory can be loosely defined a succession of transitions $(x_t, a_t, r_{t+1}, x_{t+1})$. Intuitively, consecutive transitions are temporally dependent on each other. To reflect this fact, we propose to use a structured storage method for the encoded dataset $\mathcal{Z}$. However, differently from a MDP, our transitions are *incomplete*, as we lack information of the reward. As such, in our study we refer to a transition as a tuple $(z_t, a_t, z_{t+1}, \mu_{\theta,t}(x_t), \sigma_{\theta,t}(x_t))$, where $\mu_{\theta,t}(x_t)$ and $\sigma_{\theta,t}(x_t)$ are the output vectors of the encoder of a VAE used to sample $\hat{z}_t \sim \mathcal{N}_t(\mu_{\theta,t}(x), \Sigma_{\theta,t}(x))$.

To reflect the temporal dependency across consecutive transitions and the substantial independence across trajectories, we draw inspiration from the concept of *rollout* typically used in on-policy

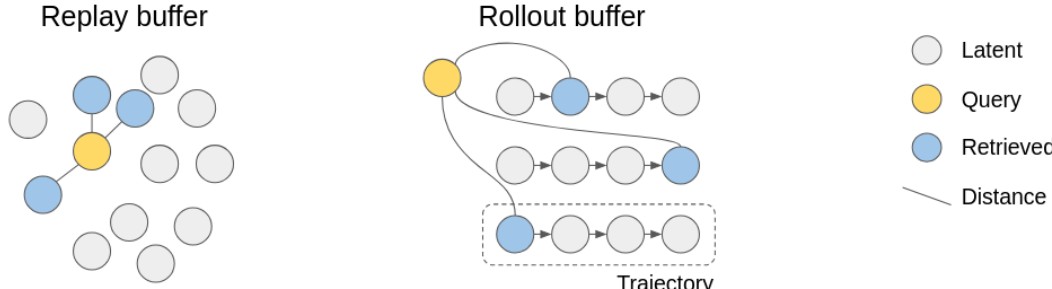

Figure 1: A simple schematic of the different data structures and sampling procedures used in our work. A replay buffer stores latents without a specific order; retrieving from it consists of computing a suitable distance of all stored data points (grey) w.r.t. the query (yellow), and retrieve the $k$-closest ones (blue). In a rollout buffer, latents are stored according to their temporal order; in this case, the search procedure only selects the 1-most similar latent from each trajectory.

RL [19, 17, 18]. In practice, we define a *rollout buffer*, specifically designed to keep each trajectory temporally ordered and independent from the others.

Given $N$ trajectories, a rollout buffer stores them in a way that generates $N$ independent *threads*; each thread has a specific order and, given a starting point in time, it can be explored only in one direction. However, multiple threads can be explored independently. In a rollout buffer, retrieval is operated as follows: given a reference latent $z_{\mathrm{ref},t}$, we retrieve the 1-most similar latent from each trajectory $\mathcal{T}$ using L2 distance and combine them in a batch of transitions $\{(\tilde{z}_{t,1}, a_{t,1}, z_{t+1,1}, \mu_{t,1}(x_t), \sigma_{t,1}(x_t)), \dots, (\tilde{z}_{t,n}, a_{t,n}, z_{t+1,n}, \mu_{t,n}(x_t), \sigma_{t,n}(x_t))\}$. Then, we extract the next state and estimate the next latent distribution as previously discussed. Similarly, conditioning the prediction of the action is immediate by restricting each search to $\mathcal{T}_{a_t} = \{(z, a, z', \mu, \sigma) \in \mathcal{T} | a = a_t\}$. In the remainder of this paper, we will refer to this method as *Rollout*.

### 3.2 Replay buffer

While a rollout buffer heuristically gives structure to the search space, we propose an simpler alternative inspired by off-policy RL [19, 14]. In detail, we convert each pair $(z_t, a_t) \in \mathcal{Z}$ in a transition $(z_t, a_t, z_{t+1}, \mu_{\phi,t}(x), \sigma_{\phi,t}(x))$ and store them in a replay buffer, which impose no constraint on ordering or temporal dependence. We propose two alternative formulations of this method, which differ in the metric used for searching. From this, other differences follow, which we detail in the next paragraphs.

**L2 distance** When searching, we retrieve the $k$-most similar transitions by computing the L2 distance between the reference latent $z_{\mathrm{ref},t}$ and each encoded $z_t$. If conditioning, we restrict the search to $\mathcal{Z}_{a_t}$. This way we extract a batch of $k$ transitions, from which we estimate $\hat{z}_{t+1}$. We name this method *Replay-L2*.

**KL divergence** Taking advantage of the vectors $\mu_\phi(x)$ and $\sigma_\phi(x)$, we propose to retrieve the most similar distribution directly, rather than estimating it from a batch of latents. To do so, given a transition $(z_t, a_t, z_{t+1}, \mu_{\phi,t}(x), \sigma_{\phi,t}(x))$, we build a reference distribution $\mathcal{N}_{\mathrm{ref}}(\mu_{\phi,t}(x), \sigma_{\phi,t}(x))$ and compute the KL divergence with all the stored distributions, extracting the index of the closest one. Then, we retrieve the subsequent transition and estimate $\hat{z}_{t+1}$ from it. Similarly to the other methods, action conditioning simply restricts the search space by masking irrelevant transitions according to their action. Perhaps unsurprisingly, we refer to this method as *Replay-KL*.

### 3.3 Comparing the methods

Intuitively, using a replay buffer gives more freedom for retrieval. However, conditioning on the action might severely impact the process, as the average distance across retrieved samples will generally increase. Hence, we expect some high variance in the retrieved batches, indicating uncertainty in

estimation. Nonetheless, each batch will produce a valid point for sampling, regardless of its semantic meaningfulness.

Searching using KL divergence solves some of the previous problems: first, since the encoded $\mu$ and $\sigma$ are obtained from actual trajectories, they correspond to meaningful regions of the latent space. Moreover, given that in this case we perform no batch estimation, we are guaranteed to sample a meaningful $\tilde{z}$. However, action conditioning might still pose a threat: whenever we diverge too much from the reference, we might get consistent, but unrelated samples.

Finally, using a rollout buffer constitutes a middle point between the other two alternatives: by imposing independency between trajectories and by sampling from each of them, we intuitively reduce the variance in the latents batch. However, if there are no similar transitions, or if a transition with the imposed action is temporally too distant from the reference, we might still observe hallucinations.

## 4 Experiments

We test our approach against PlaNet [8]. We justify the choice of using a legacy method from the Dreamer family as comparison by highlighting the good properties of this model. First, PlaNet uses a standard VAE modeled as a Normal distribution; conversely, other models of the family, starting from DreamerV2 [9], model the latent space after a discrete distribution, which may be harder to study. We remark that in our study we assume a stochastic representation, but impose no condition on a specific one. As such, using a Normal representation comes with no loss of generality. Second, PlaNet includes less modules than its evolutions. As such, observing the isolated effect of the dynamics predictor is easier.

In each experiment we compare four models, namely the baseline PlaNet along with the three versions of our approach as detailed in Section 3. We test the models on a range of visually different, image-based environments extracted from well-known benchmarks in RL. Specifically, we use five tracks from SuperTuxKart, a racing game with complex visuals, to test the performance of WMs on consistent visuals with very diverse features; two tasks from Minecraft [6], to benchmark tasks with a seemingly limitless, diverse observation space; and two tasks from Atari [14], to study how WMs predict the dynamics of small details. All our models are trained on consumer hardware, consisting of a single RTX 4080 GPU.

For each experiment, we compare the models both in latent and image spaces, using KL divergence for the first, and L1 distance and structural similarity (SSIM) for the decoded images. In particular, we use the KL divergence to assess how well the dynamics of the tasks have been reconstructed; conversely, we use MSE and SSIM to determine the visual relevance of the reconstructed dynamics. We highlight that, despite our best efforts, a VAE-reconstructed image will inevitably lose quality with respect to the original. Similarly, the KL divergence assesses the overall distance between a pair of distributions, but does not consider positional differences. As such, we invite the reader to consider the measures as correlated, and to consider the absolute values of one in light of the others.

We test our approach on both single-step and long-term predictions, using a horizon of $t = 20$ as reference for the latter. For single-step experiments, we refresh the hidden state of each model at every timestep, using the actual observation. For long-term comparisons, we let the model evolve independently until the horizon is reached, using the generated latents as intermediate representations. In both cases, we extract 20 random starting samples from a separate batch of unseen trajectories and average each measure over them.

Furthermore, since our approach uses a number of encoded trajectories for retrieval and prediction, we conduct an ablations study on two track of SuperTuxKart. We test the performance of our model for 5, 6, 7, 8, 9, 10, 15, 20 & 30 encoded trajectories. To establish a fair comparison, we train a SSM for each subset of data, using the same trajectories in both cases. Similarly to other experiments, we extract 20 random samples from a disjoint set of test trajectories and report the average for each measure.

Finally, we recall how our approach enables us to estimate $\mathbb{P}(s_{t+\Delta t}|s_t)$ with $t \in \mathbb{N}, \Delta t > 0$ as well, that is, how the state is expected to evolve from the current one with on restriction on the chosen action. Since we deem this difference interesting, we compare our models when acting with and without action conditioning.

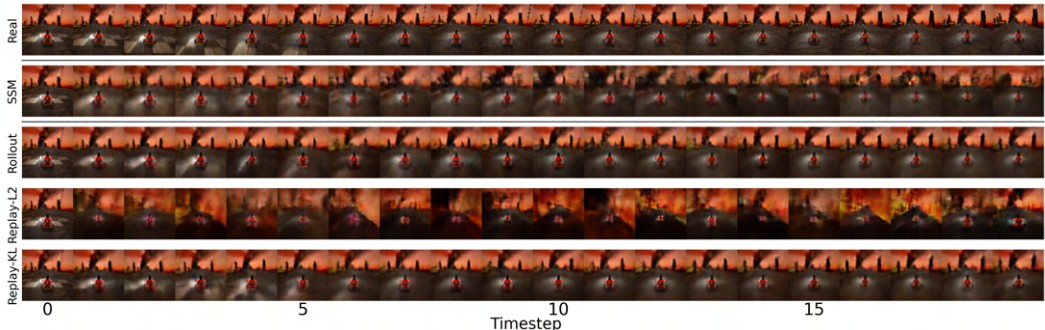

Figure 2: Long-horizon reconstruction of 20 consecutive frames in SuperTuxKart. The first row shows the real sampled sequence of frames. Each row from second to last corresponds to a model, respectively, a SSM baseline, a search-based world model with each trajectory encoded as a separate rollout, an L2-search-based world model with no constraint on trajectories, and a KL-search-based world model without temporal constraints. To predict the next frame, each model can only use the predicted context from the previous timestep.

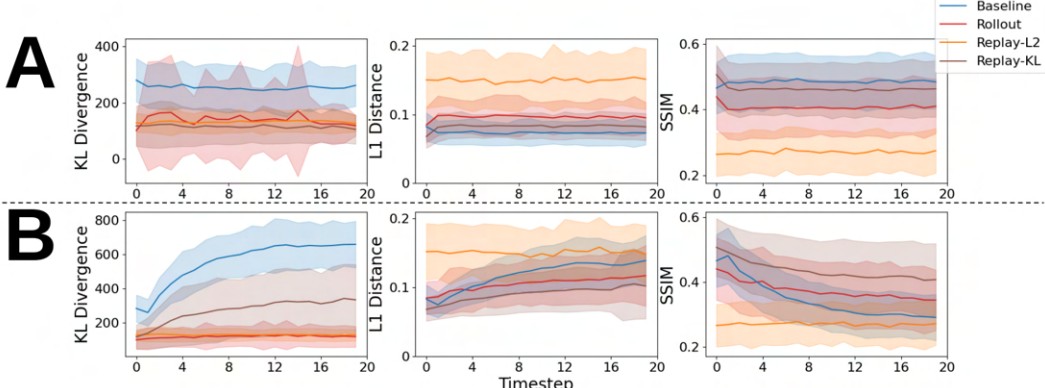

Figure 3: Numerical comparison of the models on the SuperTuxKart reconstruction benchmark for (A) one-step and (B) long-horizon predictions. Values are computed by averaging over five tracks, selecting 20 random images from a disjoint set of test trajectories. For each value, we report mean and variance at each timestep, up to the horizon fixed at $t = 20$.

## 5 Results & Discussion

In this Section, we present the results of our study. The Section is organized in three subsections, each dedicated to a specific experiment as described in Section 4. Due to page limit constraints, we report only a small subset of the available results and present only a handful examples. However, we report more details and examples in Appendices.

### 5.1 Prediction & reconstruction quality

Figure 2 shows an example of long-horizon dynamics prediction using each model. In the one-step case, we see how re-initializing the hidden state of the model benefits their performance. We highlight how PlaNet hallucinates after 10 timesteps, while Rollout and Replay-KL maintain consistency in their prediction. In particular, Replay-KL shows an unmatched resemblance to the real sequence. In both cases, Replay-L2 is affected by noticeable hallucinations, thus making it completely unreliable. In the supplementary material, we include more examples from different tracks, and examples of one-step predictions.

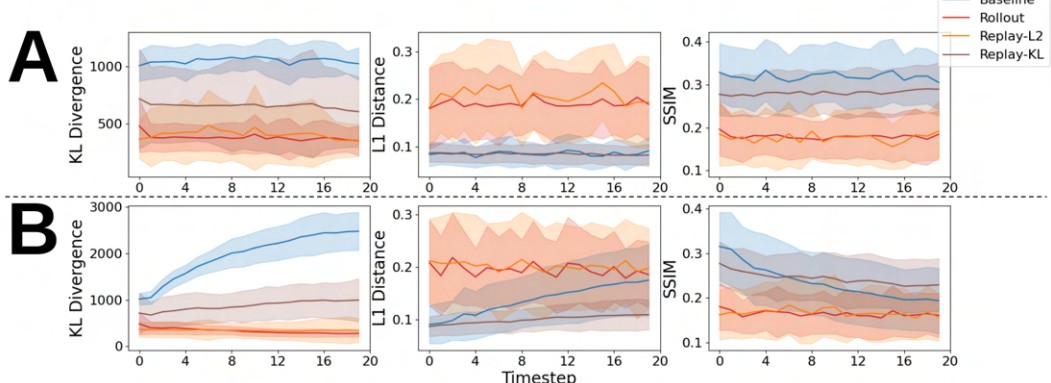

Figure 4: Cumulative benchmark for (A) one-step and (B) long-horizon predictions in Minecraft. The values are averaged over two tasks, "Treechop-v0" and "Navigate-v0". For each task, we select 20 random transitions from the test dataset and report the evolution of KL divergence, L1 distance and structural similarity over time.

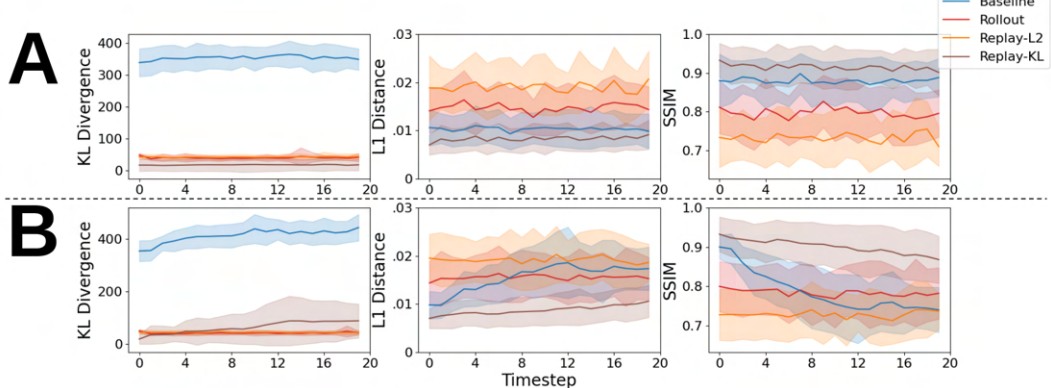

Figure 5: Error measures over two tasks of the Atari benchmark, separated in (A) one-step and (B) long-horizon. In our experiments, we use "Seaquest" and "Space Invaders" to explore the limitations of our models in reconstructing small details. For each task, we average over 20 random transitions of the test set and reconstruct for $t = 20$ timesteps.

The numerical comparison reported in Figure 3 confirms our qualitative assessment: for one-step predictions (Figure 3A) the baseline, Rollout and Replay-KL are visually indistinguishable, while in long-horizon regime Rollout and especially Replay-KL achieve remarkable performance over the baseline. It is notable how all our models are generally closer in distribution to the real representation than the baseline. We explain this mismatch by remarking that a VAE-reconstructed image will necessarily carry some error. As such, comparison between latent reconstructions and decoded images are not trivial.

Notably, Replay-L2 features a small KL error, but significantly higher L1 and lower SSIM values. We explain this fact as follows: searching in a replay buffer while conditioning on an action may lead to high variance in the retrieved batch. Hence, estimating the posterior from it could lead to an unstable region of the latent space. Therefore, sampling in that point may produce visually incoherent reconstructions. On the other hand, giving structure to the latent space through the separation in rollouts appear to regularize the reconstruction of $\hat{x}_{t+1} \sim p(\boldsymbol{X_{t+1}}|\hat{z}_{t+1})$. As for Replay-KL, searching directly for the most similar prior greatly reduces this instability. Hence, in both cases we have more coherent predictions.

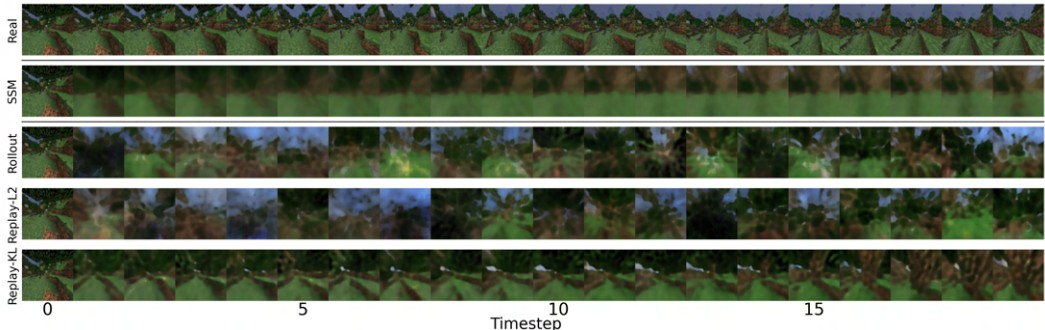

Figure 6: Long-horizon reconstruction from Minecraft "Treechop" task. First row reports the real sequence; rows from second to last show the reconstruction of, respectively, baseline, Rollout, Replay-L2, and Replay-KL.

We report the reconstruction results for Minecraft and Atari respectively in Figures 4 & 5. We leverage these two environments to benchmark specific aspects of the compared models, namely the ability to reconstruct in a seemingly infinite observation space, and the ability to focus on small details. In both cases, we confirm the patterns discussed for SuperTuxKart: Rollout and Replay-KL are generally better than the SSM baseline, while Replay-L2 is worse than any other model in terms of actual reconstruction, while remaining overall competitive in latent space.

By comparing values in Figures 3, 4 & 5 we conclude that, as expected, Minecraft represents a challenging environment for reconstruction: despite the similar patterns, we see how values are overall higher across all measures. In particular, Rollout and Replay-L2 behave similarly both in terms of dynamics prediction and reconstruction. However, the example shown in Figure 6 shows how both models produce a completely unreliable reconstruction.

Despite obtaining similar values in L1 and SSIM, the baseline and Replay-KL produce very dissimilar results: on one hand, the SSM baseline produces blurry, incoherent images; on the other hand, Replay-KL generates visually appealing sequences that recall the main elements of a scene, but gradually diverges. By isually analyzing the samples, we notice a "hit-or-miss" pattern in both models, with Replay-KL being generally sharper but more variable, and the baseline being much blurrier but more accurate in overall structure. In the supplementary material of this study we report some examples of this fact and offer a more detailed analysis to support this claim.

## 5.2 Ablation study: number of trajectories

We report the results of our ablation study on SuperTuxKart in Figure 7, and provide an additional ablation on Minecraft in supplementary material. As expected, the SSM baseline benefits from the greater availability of training data; on the other hand, increasing the dataset only marginally improves our models, and only in some cases.

In detail, KL divergence (Figure 7) in the SSM baseline drops steadily from $339.48$ to $169.46$, amounting to a total decrease of $50.08\%$, in the one-step case. In long-horizon predictions, the decrease is less consistent ($-26.56\%$). Similarly, Rollout benefits from more data and reduces the average KL divergence from $311.89$ to a mere $86.27$ ($-72.34\%$). In contrast, Replay-L2 and Replay-KL are substantially unaffected by the increase in available data. We explain this fact by noting that the result of an unconstrained search changes only if there is no meaningful data. As for Rollout, we point out that introducing a temporal constraint on the encoded trajectories severely impacts search, and adding more data seems relax this constraint. However, if we consider the absolute values for each model, we see how the best baseline model (30 trajectories) still reports higher average KL than Rollout and Replay-KL with only five encoded trajectories.

L1 distance and SSIM substantially confirm our previous discussion: decodings from the SSM baseline improve with more data, while Replay-KL and Rollout remain substantially unaffected, and perform either comparably or better than the baseline in all cases. In the Appendices we show some image reconstructions directly related to these results.

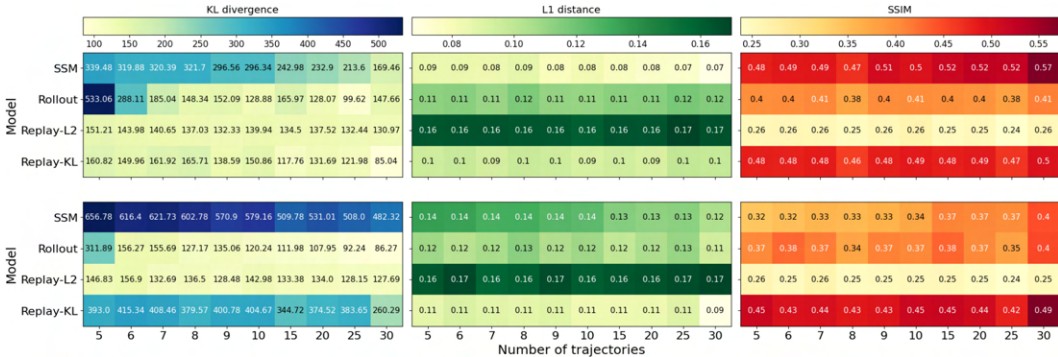

Figure 7: Results of the ablation study on two tracks from SuperTuxKart. Each model uses an increasing number of trajectories for either training (SSM) or retrieval (Rollout, Replay-L2, Replay-KL). Top row shows results of one-step predictions, bottom row corresponds to long-horizon dynamics reconstruction. Heatmaps analyze (from left to right) KL divergence, L1 distance, and structural similarity.

Table 1: Performance for planning in SuperTuxKart. Results are reported using mean reward & standard deviation, and a 95% confidence interval (parentheses).

| Model | fortmagma | lighthouse | snes_rainbowroad | snowmountain | volcano_island |
|---|---|---|---|---|---|
| PlaNet | $0.0880 \pm 0.1125$ (0.0557, 0.1203) | $3.0780 \pm 1.5162$ (2.6427, 3.5133) | $3.1600 \pm 2.9678$ (2.3080, 4.0120) | $5.3220 \pm 3.7877$ (4.2346, 6.4094) | $0.9280 \pm 0.6440$ (0.7432, 1.1128) |
| Rollout | $\mathbf{5.8100 \pm 3.4761}$ **(4.8121, 6.8079)** | $2.6400 \pm 2.5952$ (1.8950, 3.3850) | $4.8220 \pm 1.1188$ (4.5008, 5.1432) | $5.5560 \pm 6.5701$ (3.6698, 7.4422) | $1.5300 \pm 0.5849$ (1.3621, 1.6979) |
| Replay-L2 | $4.3500 \pm 1.4506$ (3.9336, 4.7664) | $\mathbf{5.1140 \pm 9.5549}$ **(2.3709, 7.8571)** | $\mathbf{6.3360 \pm 5.2006}$ **(4.8430, 7.8290)** | $4.9560 \pm 6.5035$ (3.0890, 6.8230) | $3.3480 \pm 6.1291$ (1.5885, 5.1075) |
| Replay-KL | $4.7740 \pm 3.8107$ (3.6800, 5.8680) | $0.3200 \pm 0.7558$ (0.1030, 0.5370) | $4.7140 \pm 0.6515$ (4.5270, 4.9010) | $\mathbf{15.6640 \pm 11.8248}$ **(12.2693, 19.0587)** | $\mathbf{4.4780 \pm 7.9652}$ **(2.1913, 6.7647)** |

## 5.3 Action selection

WMs are typically embedded in more complex architectures to inform the state with future information. For instance, our baseline model PlaNet [8] is designed for *planning*, that is, selecting a sequence of actions by evolving the environment dynamics from the current observation. Although not the primary scope of this paper, we deemed it interesting to explore the planning capabilities of our methods.

We test each model on the realized returns over 50 evaluation episodes using the SuperTuxKart benchmark, as we believe that it represents the typical use case for our approach. Each model is allowed to observe the first state of an episode. From this single observation, agents are asked to evolve the dynamics according to their WM and plan the next 20 actions. Then, the planned actions are executed with no room for corrections; finally, the model is allowed to observe a new state and plan the next sequence of actions. The process repeats until the end of the episode.

Actions are selected deterministically using the mode of a retrieved action distribution. We report the results in Table 1, highlighting the best model for each environment. For each track, we report mean, standard deviation, and 95% confidence interval.

The results clearly show that our approaches perform generally better than the PlaNet baseline, thus suggesting that search-based approaches *can* plan. In particular, replay buffer-based methods yield significant improvements in 4 out of 5 environments. However, from our experiments, no specific method clearly dominates on the others.

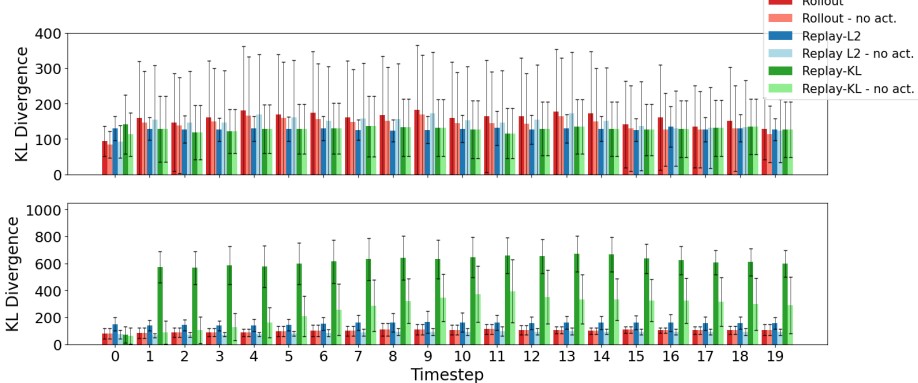

Figure 8: Evolution of KL divergence w.r.t. action conditioning for (Top) one-step and (Bottom) long-horizon predictions. A reference model is reported using solid color; corresponding models without action conditioning are characterized by a pastel tone of the same color.

## 5.4 Action conditioning

Figure 8 shows the results of our study on the effects of action conditioning. We remark that the scope of this experiment is to assess the differences between predicting $\mathbb{P}(z_{t+1}|z_t, a_t)$ and $\mathbb{P}(z_{t+1}|z_t)$. Since an SSM can not compute the latter, we only compare Rollout, Replay-L2, and Replay-KL.

The results confirm the generally poor performance of Replay-L2: both in one-step (Figure 8A, top) and long-term (Figure 8, bottom) predictions, KL divergence significantly decreases. This indicates a reduction in variance over the batch of retrieved samples, consistently with our speculation. However, by analyzing the decoded sequences related to this study (reported in Appendix B), results are still too noisy to be considered reliable.

Rollout marginally benefits from the absence of actions in one-step predictions, while apparently remains unaffected in the long-horizon regime. Analyzing the decoded sequences (in Appendix B) reveals no substantial difference. We explain this fact in conjunction with the result obtained in Section 5.2: intuitively, predicting $\mathbb{P}(z_{t+1}|z_t)$ rather than $\mathbb{P}(z_{t+1}|z_t, a_t)$ enlarges the pool of retrievable images.

In Replay-KL, one-step predictions are unaffected by action conditioning. Contrarily, long-term reconstructions show a significant increase in KL divergence for conditioned retrieval. We explain this effect by highlighting that Replay-KL recovers only one example for each search and, in the long-term scenario, only retrieves the first latent. Forcing the model to take a specific action may rarely result in planning on a very dissimilar sequence of latents.

However, by considering the one-step case, where the same retrieval produces virtually no effect, this search failure appears to be a very rare occurrence. Moreover, we believe that the best use for our method lies in the "no action conditioning" regime. Additionally, we do not deem this error to be a serious limitation: intuitively, enlarging the data pool in memory is likely to mitigate this effect.

## 6 Conclusions

We have presented an alternative way to predict the transition dynamics on a number of image-based environments. Our proposal explores similarity search on stochastic representations as a way to build WMs without training. Our results confirm that our proposed WM can act as a valid alternative in a range of scenarios.

Our methods outperform the baseline while also being more immediate in implementation and application. Despite showing acceptable performance in open-ended tasks such as Minecraft and being comparable to the PlaNet baseline, our results suggest that the performance of zero-shot WMs depends on latent space coverage of the encoded trajectories, limiting their effectiveness in tasks with vast state spaces. However, they represent a valid, lightweight alternative to regular WMs in small-scoped tasks, such as SuperTuxKart and Atari.

**Acknowledgments.** The authors wish to acknowledge CSC – IT Center for Science, Finland, for computational resources. Additionally, Ville Hautamäki thanks the Jane and Aatos Erkko Foundation for partial funding.

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

# A Prediction & reconstruction - benchmark results

In this Appendix we report the evaluations for each environment we tested. We report the results following the same template as Figure 3, 4 and 5. Additionally, we discuss each result individually, to better highlight environment-related strengths and weaknesses of each approach.

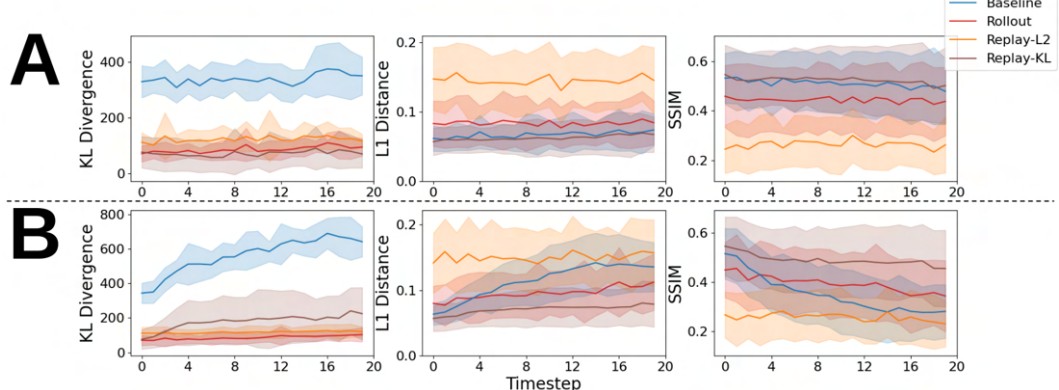

Figure 9: Numerical evaluation in SuperTuxKart "fortmagma" track. Results are reported in terms of KL divergence for latent dynamics prediction, and L1 distance & SSIM index for (A) one-step and (B) long-horizon predictions.

Results for the "fortmagma" (Figure 9) reflect the general behavior we have discussed in Section 5: all our models achieve significantly lower error in KL divergence, meaning that they are closer to the true dynamics in latent space. As for the reconstruction, which we remark could be affected also by an error in decoding the latent, Rollout and Replay-KL show comparable reconstructions with the baseline, while Replay-L2 is generally affected by hallucinations or inconsistencies.

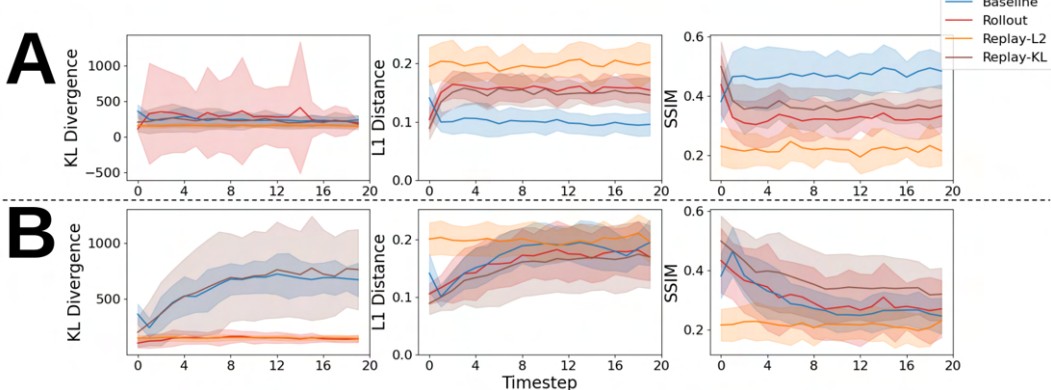

Figure 10: Numerical evaluation in SuperTuxKart "snes_rainbowroad" track. Results are reported in terms of KL divergence for latent dynamics prediction, and L1 distance & SSIM index for (A) one-step and (B) long-horizon predictions.

The results of our benchmark on "snes_rainbowroad", reported in Figure 10, show the importance of interpreting the quantitative and qualitative results jointly. The general behavior of the models is quite different from Figure 9: our models Rollout and Replay-KL perform either similarly (KL divergence) or slightly worse (L1, SSIM) than the baseline for one-step predictions, while match or improve in long-term dynamics prediction.

We attribute this difference to the visual complexity of the images from this specific environment, as shown in Appendix B: the sequence of colors in the track, along with the "falls" that the player might

experience heavily contribute to the divergence from the real sequence. However, we point out how our models are generally more consistent in reconstructing visually coherent sequences, even though they might be slightly more distant in terms of visual accuracy.

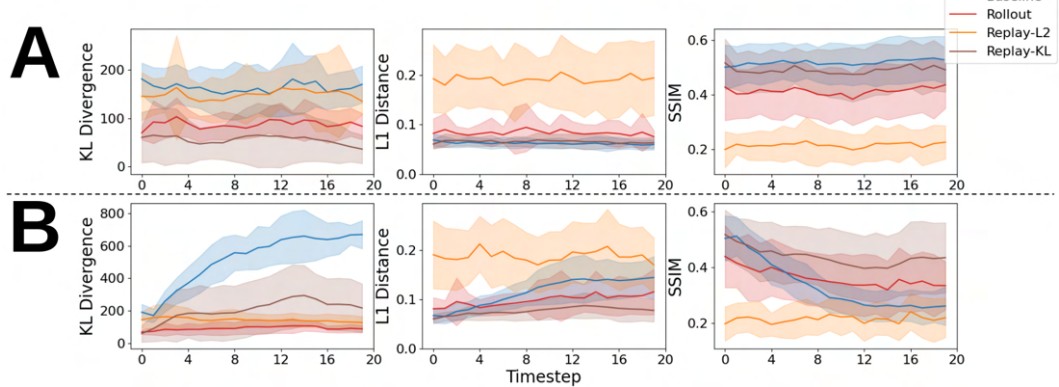

Figure 11: Numerical evaluation in SuperTuxKart "volcano_island" track. Results are reported in terms of KL divergence for latent dynamics prediction, and L1 distance & SSIM index for (A) one-step and (B) long-horizon predictions.

The results from track "volcano_island" (Figure 11) support the general results we have shown in the main text. Our agents Rollout ad Replay-KL are generally closer to the real sequence in terms of latent dynamics prediction, while either match or slightly improve the visual reconstructions. Notably, the gap in difference is more pronounced in long-horizon predictions, where Rollout and Replay-KL diverge from the real sequence much more slowly than the baseline.

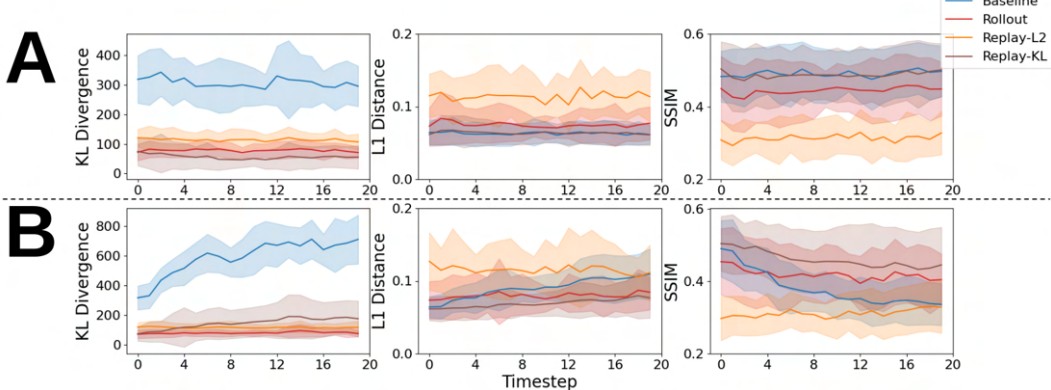

Figure 12: Numerical evaluation in SuperTuxKart "snowmountain" track. Results are reported in terms of KL divergence for latent dynamics prediction, and L1 distance & SSIM index for (A) one-step and (B) long-horizon predictions.

In the "snowmountain" track, the overall trend is also confirmed: all our models improve latent reconstructions, while either match or improve the baseline in decoded reconstructions. The only exception is represented by Replay-L2 that, despite achieving a better latent prediction, systematically fails to decode it. As reported in main text, we track this effect to the latent space learned by the VAE: while estimating the generative distributions from a batch of similar samples correctly points the model in the right direction, sampling from that distribution for the purpose of decoding usually results in visually unpleasant images.

As can be seen from the examples in B, "lighthouse" features a visually dark theme. As such, differences in this track are generally smaller and harder to assess from quantitative measures.

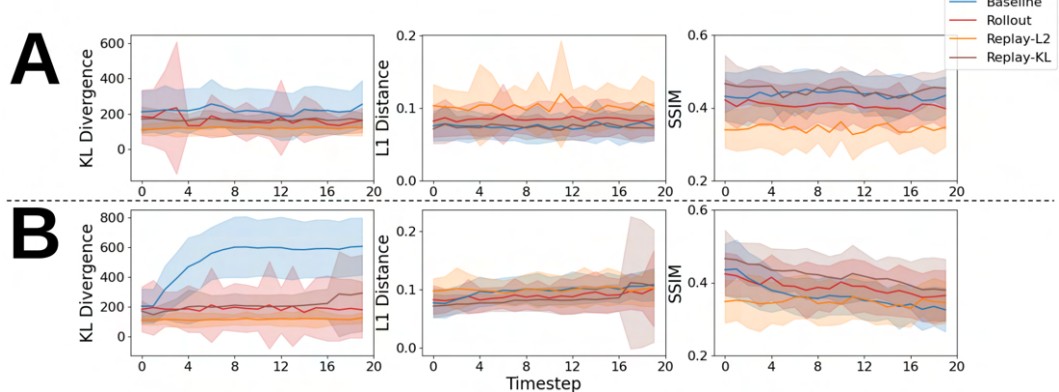

Figure 13: Numerical evaluation in SuperTuxKart "lighthouse" track. Results are reported in terms of KL divergence for latent dynamics prediction, and L1 distance & SSIM index for (A) one-step and (B) long-horizon predictions.

However, we see from Figure 13 how the general trend is confirmed, with Replay-KL and Rollout either matching or improving over the baseline, while Replay-L2 behaving better than the baseline in latent reconstructions (KL divergence), while failing to properly decode its predictions.

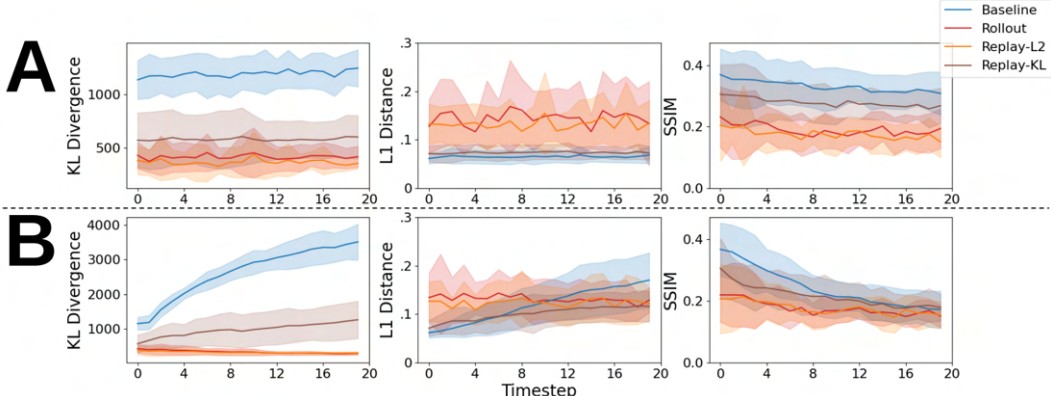

Figure 14: Numerical comparison for Minecraft "Treechop-v0" task. Results are divided in (A) one-step and (B) long-horizon predictions.

The results from "Treechop-v0" (Figure 14 from the Minecraft benchmark further confirm our findings. Intuitively, the state space of the task is limitless, with a few "bottlenecks" occurring in the proximity of a tree. Nonetheless, our agents are generally better at reconstructing latent dynamics, and on average match the baseline on visual reconstructions. We found this result surprising, as our agents only store a limited number of trajectories, while an SSM should theoretically be able to generalize better.

Similarly, in "Navigate-v0" a human expert is asked to explore the open world of the game, with no particular aim. Hence, it is nearly impossible to fully represent the full space without having access to an infinite number of trajectories. The results in Figure 15 reflect this intrinsic complexity: this is the only instance in our experiments where the latent reconstruction favors the baseline over our strongest model, Replay-KL. However, we notice how, despite similar or higher KL divergence, our models' decoded latents are generally closer to the real sequence.

Our experiments on "Space Invaders", reported in Figure 16 reflect the average trend shown in the main text. Notably, in this case the KL divergence computed on our models is very low, meaning that the predicted dynamics is generally very close to the original sequence. Also, due to the images

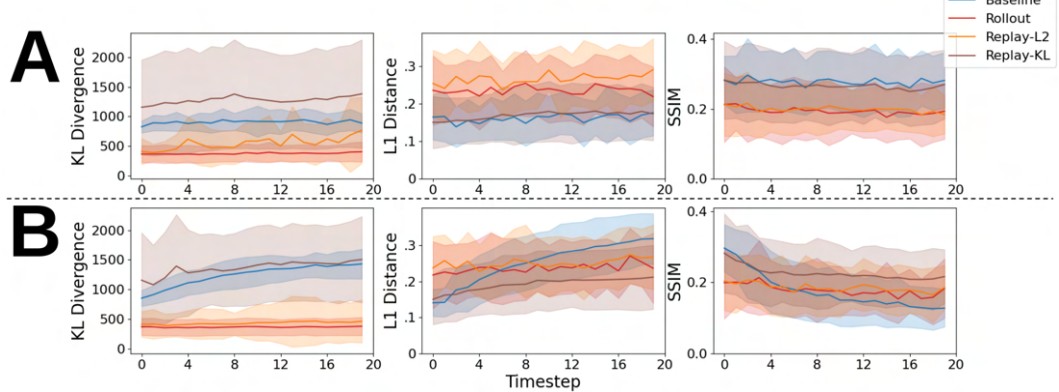

Figure 15: Numerical comparison for Minecraft "Navigate-v0" task. Results are divided in (A) one-step and (B) long-horizon predictions.

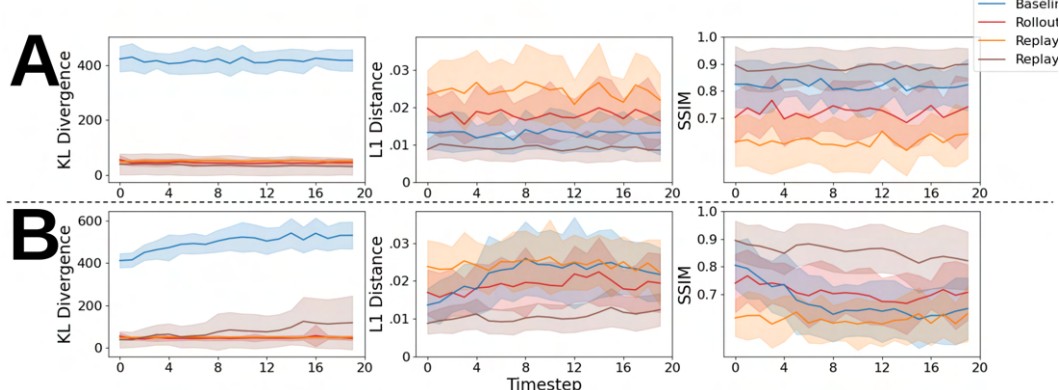

Figure 16: Numerical evaluation of "Space Invaders" from the Atari benchmark. Results report (A) one-step and (B) long-horizon predictions.

differing only on small details, the error on L1 distance is an order of magnitude lower w.r.t. the other benchmarks, and SSIM is close to perfect.

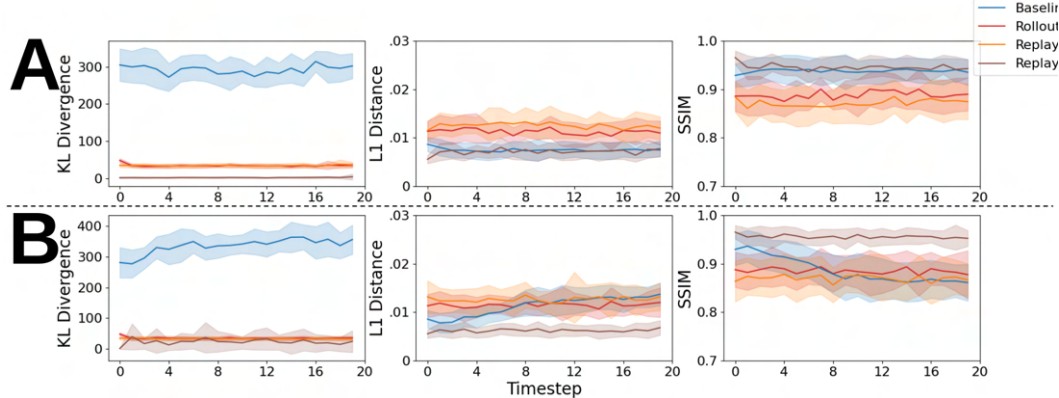

Figure 17: Numerical evaluation of "Seaquest" from the Atari benchmark. Results report (A) one-step and (B) long-horizon predictions.

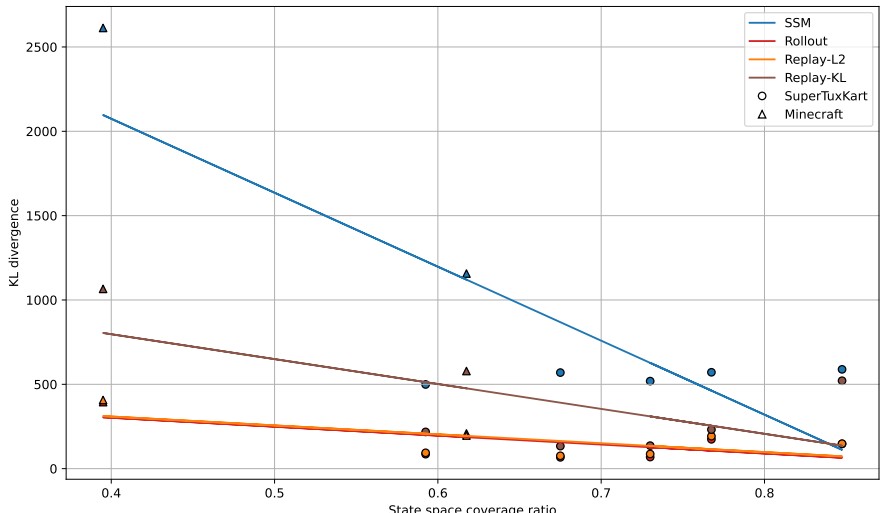

Figure 18: A comparison between latent space coverage ratio, that is, the fraction of latent space "sectors" covered by the encoded trajectories, and the KL prediction error of each world model.

Table 2: Pearson correlation index of the tested models representing the correlation between encoded trajectories and KL prediction error.

| Model | Pearson Correlation |
|---|---|
| PlaNet | -0.8269 |
| Rollout | -0.6709 |
| Replay-L2 | -0.6711 |
| Replay-KL | -0.6366 |

Similarly, Figure 17 further confirms the validity of our approach, with Replay-KL performing either on par or better than the baseline. Moreover, like in "Space Invaders", the gap in KL divergence is abyssal, while L1 distance and SSIM are similar, but always favoring Replay-KL, especially in long-horizon reconstructions.

## A.1 Coverage statistics

In Section 6, we concluded that our methods are most expressive in smaller-scoped tasks, in which the set of encoded trajectories are sufficient representations of the task dynamics. However, due to page limitations, no empirical result was provided. In this Section, we test the generalization capabilities of our proposed approaches w.r.t. to the PlaNet baseline. To do so, we compare the latent state coverage ratio of the encoded trajectories with the KL prediction error of each model. Additionally, we compute the Pearson's correlation index for each tested environment and model. The results are shown in Figure 18 and Table 2

The experiment shows that, as denoted in Section 6, higher coverage of the latent space corresponds to a lower prediction error. This supports our claim that our models are better suited for smaller scoped tasks, such as SuperTuxKart and Atari, where a significant fraction of the environment dynamics is implicitly reproduced in the encoded trajectories. However, interestingly the strongest correlation is observed in the PlaNet baseline, indicating that both learning and search-based world models benefit similarly from more data.

## B   Prediction & reconstruction - qualitative evaluation

This Appendix reports a number of visual reconstructions of predicted dynamics. For each environment, we report two one-step and two long-horizon predictions, namely a positive example and a failure case for our models. Given the number of models we test, finding a totally positive/negative

example that include all models is hard. As such, in order to maintain the length of our Appendix reasonable, we report the *best overall* positive/negative examples. Each subsection refers to an environment.

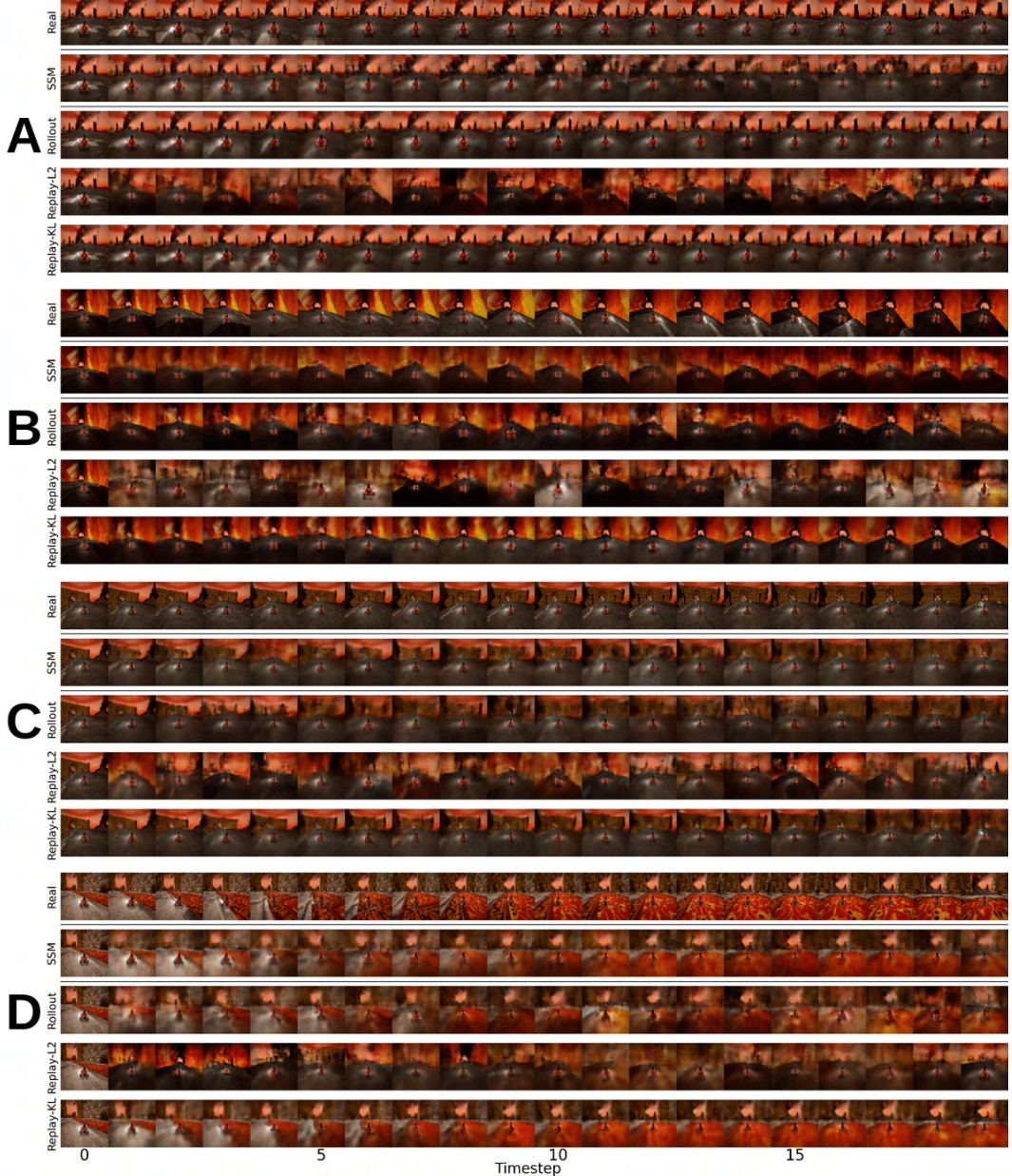

Figure 19: Examples from SuperTuxKart "fortmagma" track. (A) Long-horizon success; (B) long-horizon failure; (C) One-step success; (D) One-step failure.

In "fortmagma" (Figure 19) we could not find a failure case for Replay-KL. However, Rollout and especiallt Replay-L2 are subject to hallucinations both in one-step and long-term reconstruction.

As stated in the main text, "snes_rainbowroad" represents a particularly hard case for this benchmark due to the visual complexity of the track and to the "falls" that a player may experience. As such, in Figure 20 we see how all models can incur into hallucinations.

The track "volcano_island" features a lot of visual variety, as it includes track sections on asphalt, rock, sand, and even water. Such visual diversity generally makes prediction harder. As expected,

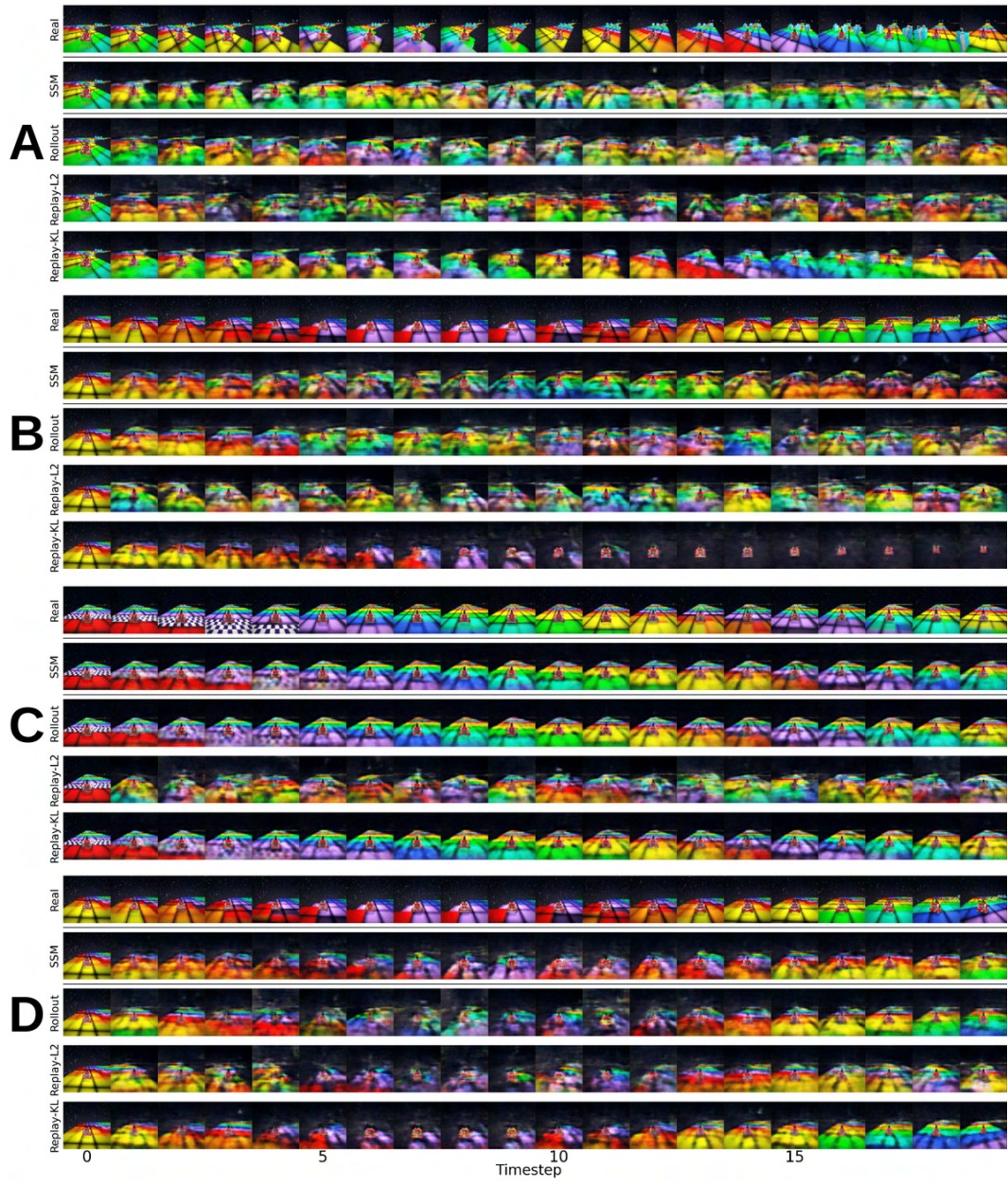

Figure 20: Examples from SuperTuxKart "snes_rainbowroad" track. (A) Long-horizon success; (B) long-horizon failure; (C) One-step success; (D) One-step failure.

in Figure 21 we see how, to some extent, all models incur into hallucinations. An exception is represented by Replay-KL, which exhibits strong performance throughout the whole track.

As visible in Figure 22, in "snowmountain" all models except Replay-L2 are mostly capable of predicting both long-term and one-step situations. However, we see in Figure 22A that the baseline SSM model might diverge from the reference sequence.

As previously stated, "lighthouse" is a peculiar case of this benchmark due to its intrinsic darkness. As such, spotting hallucinations is a hard task. However, the tracks also features lightnings occurring at random times, which we use to determine failure cases. In particular, Figure 23B shows Replay-KL predicting a lightning when not existing. Also, in section D of the same Figure we see how all models fail to reconstruct a coherent background.

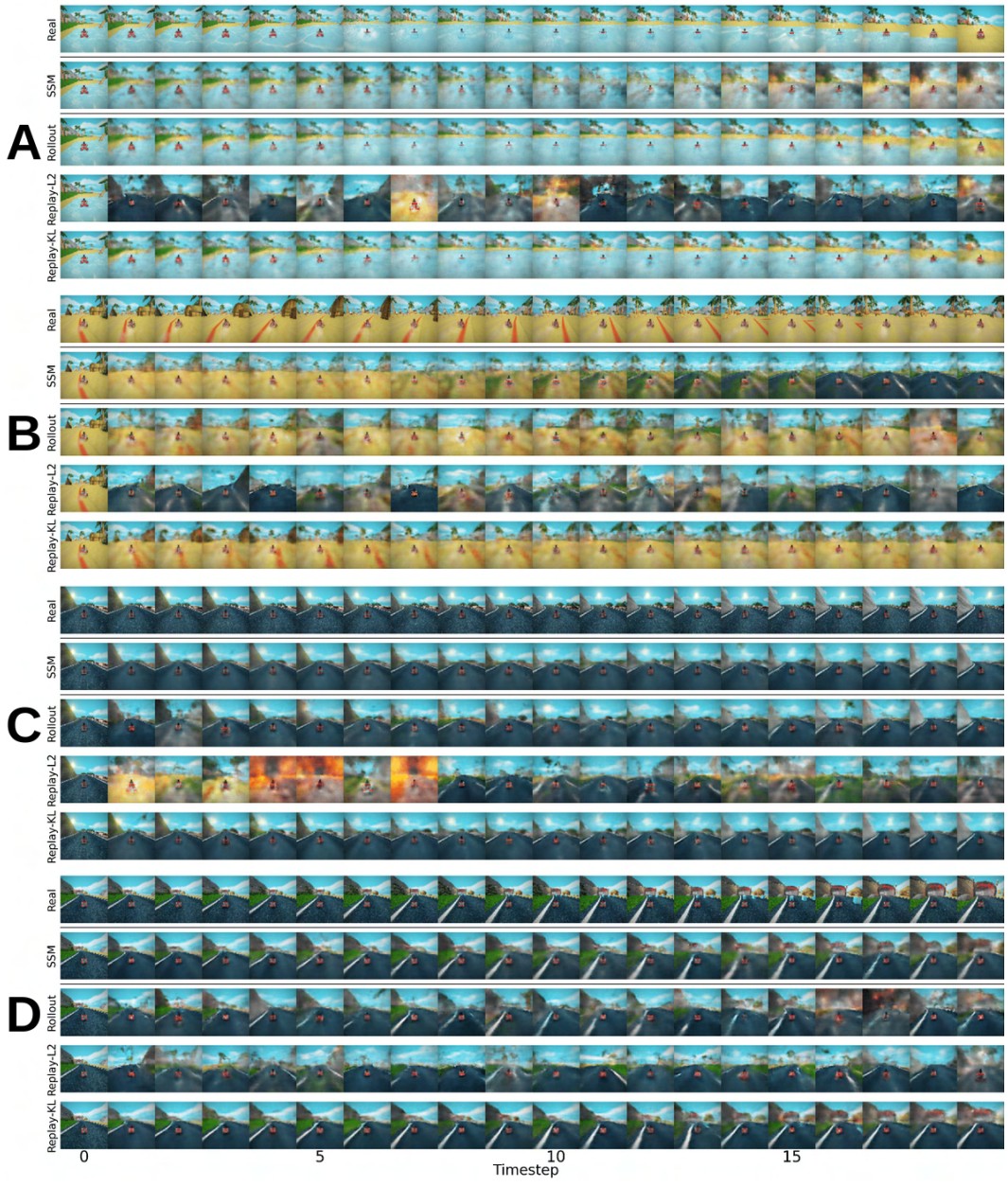

Figure 21: Examples from SuperTuxKart "volcano_island" track. (A) Long-horizon success; (B) long-horizon failure; (C) One-step success; (D) One-step failure.

Tasks from the MineRL benchmark feature an almost limitless state space. As such, expecting a perfect prediction from quite simple models such as ours is quite irrealistic. However, in Figure 24 it is surprising to see Replay-KL achieving quite good reconstructions in A and C. Notably, in this task our baseline model (SSM) produces mostly blurry reconstructions.

An even more extreme case of limitless state space is represented by "Navigate-v0", for which we report some examples in Figure 25. Interestingly, quite some trajectories are centered on navigating the sea, hence easily leading all models to hallucinations if water is shown in the picture. For this task, finding a "good prediction" was particularly hard.

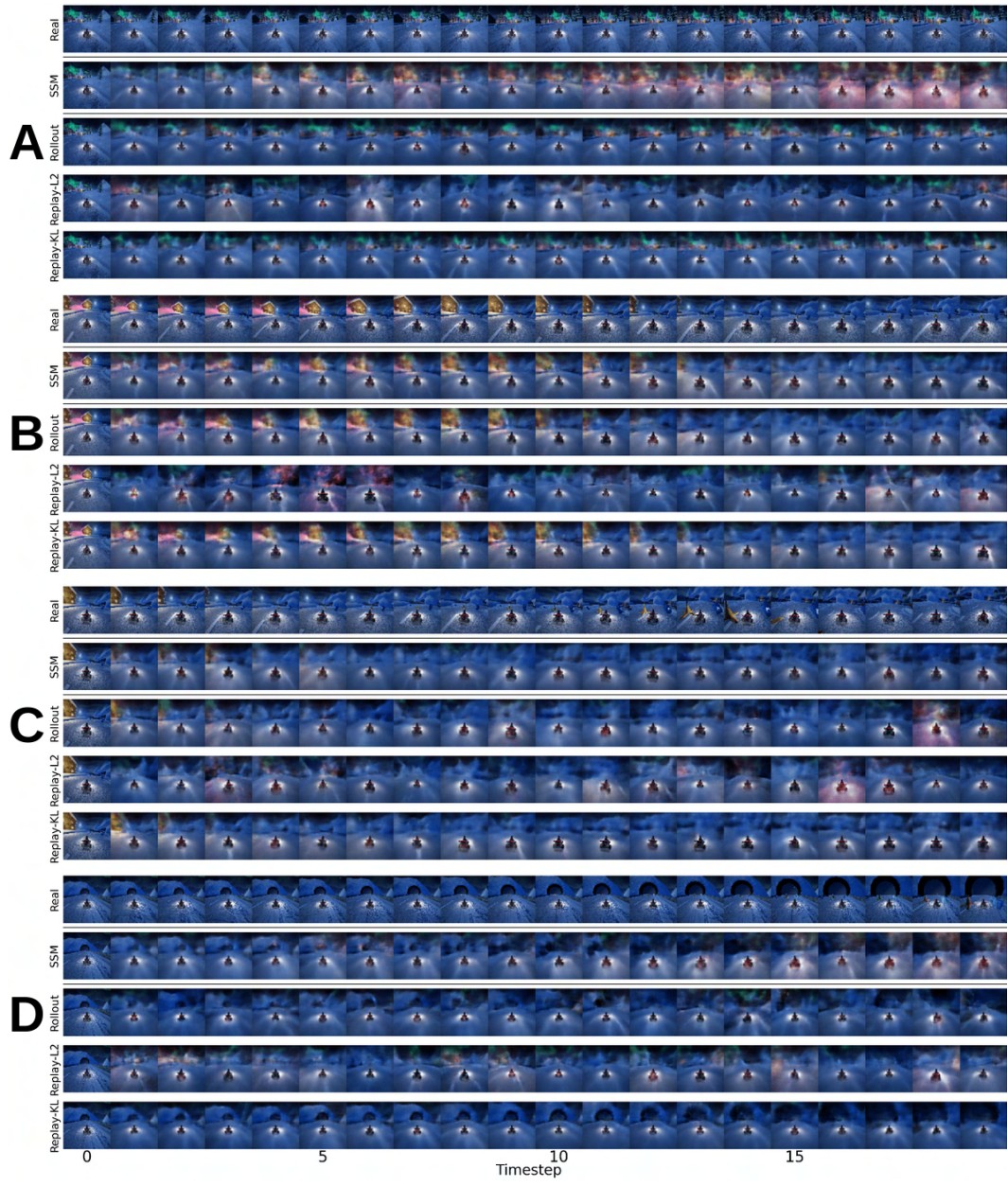

Figure 22: Examples from SuperTuxKart "snowmountain" track. (A) Long-horizon success; (B) long-horizon failure; (C) One-step success; (D) One-step failure.

The most notable difference between the models in "Space Invaders" (Figure 26 is the consistency with which they predict the alien ships. In general, Replay-KL was the most consistent one, even though it was not perfect.

Similarly, as shown in Figure 27 in "Seaquest" the most notable example of hallucination is represented by the enemy boats that, in some models, flicker throughout the sequence. Like in "fortmagma", Replay-KL was mostly correct and coherent in its predictions, while both SSM, Rollout and Replay-L2 struggle to maintain consistency in their predictions.

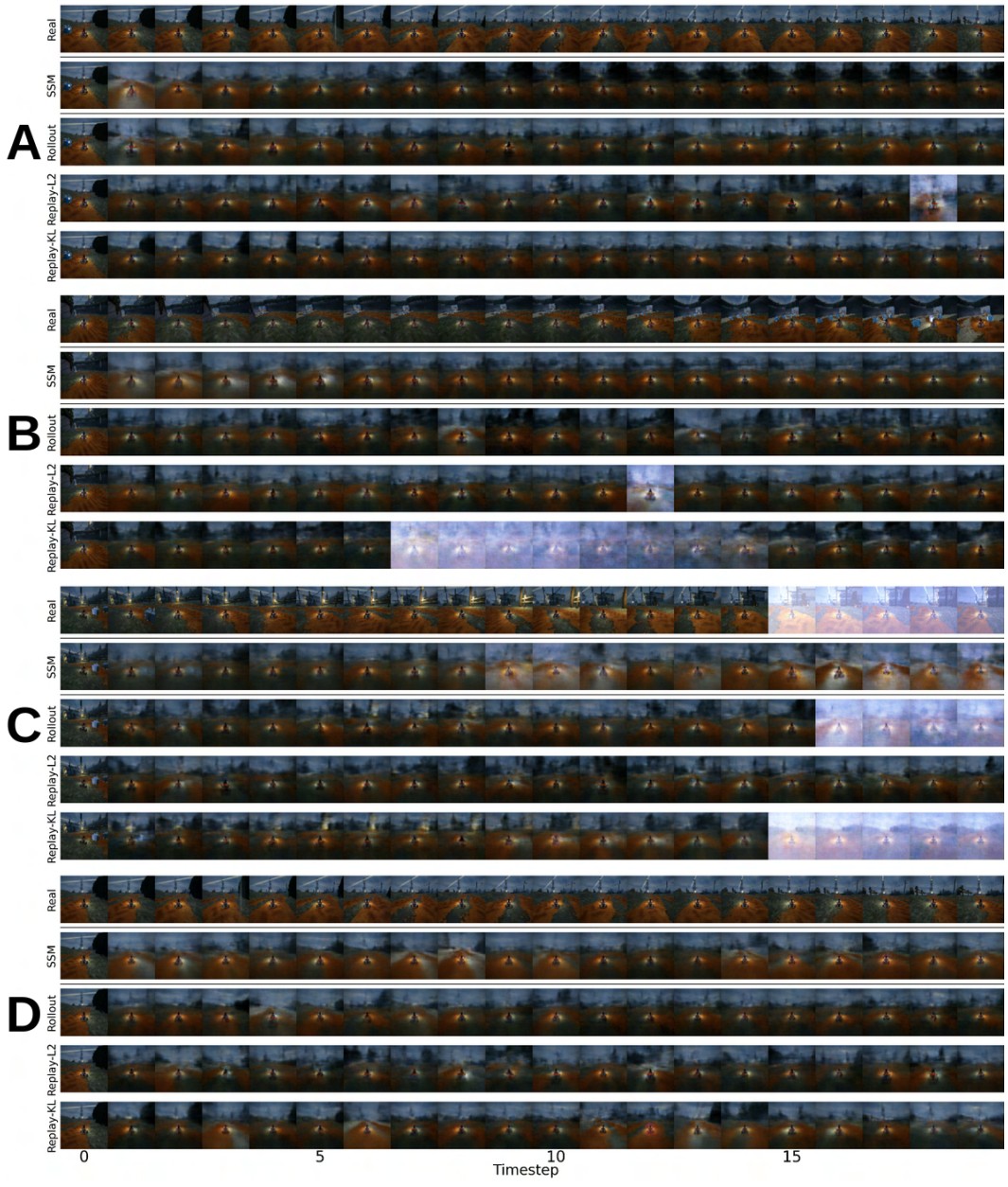

Figure 23: Examples from SuperTuxKart "lighthouse" track. (A) Long-horizon success; (B) long-horizon failure; (C) One-step success; (D) One-step failure.

# C  Ablation study - additional results

In this Appendix we report some visual reconstructions coming from the ablation study presented in Section 5.2. Moreover, we report the results of an additional ablation study performed on Minecraft "Treechop-v0", using $[10, 80]$ trajectories and a step of 10. This additional ablation study is also used to compute the wallclock time needed for each method to predict the latent dynamics. To fit all the models in one run, the test is run on CPU, using an Intel i7 12650HX.

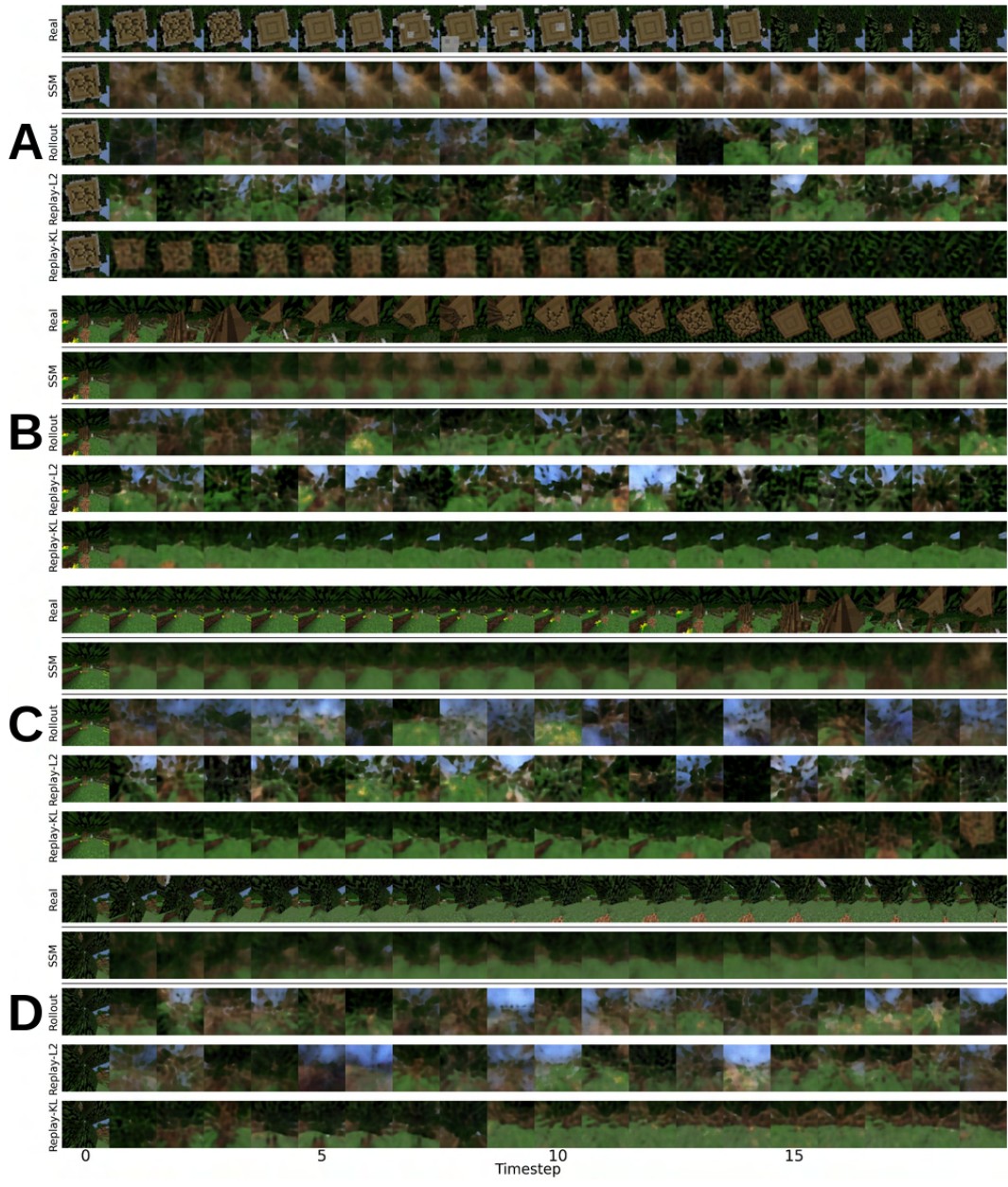

Figure 24: Examples from MineRL "Treechop-v0" task. (A) Long-horizon success; (B) long-horizon failure; (C) One-step success; (D) One-step failure.

## C.1 Wallclock time

Figure 28 shows the average time required to recover a transition for each method in the one-step (top) and long-horizon (bottom) cases. As expected, the time required to perform the search scales linearly in the number of stored transitions. During the test, we occupied a maximum of 12GB of RAM, including VAE, SSMs, a rollout and a replay buffer.

Clearly, computing the KL divergence requires significantly more time than computing L2 distance. However, we highlight that for long-term predictions we only require to compute the distances once every $N$ steps, where $N$ is the horizon. Additionally, we inform that in our implementation we used the `torch.distributions` package to ensure correctness, regardless of efficiency. The

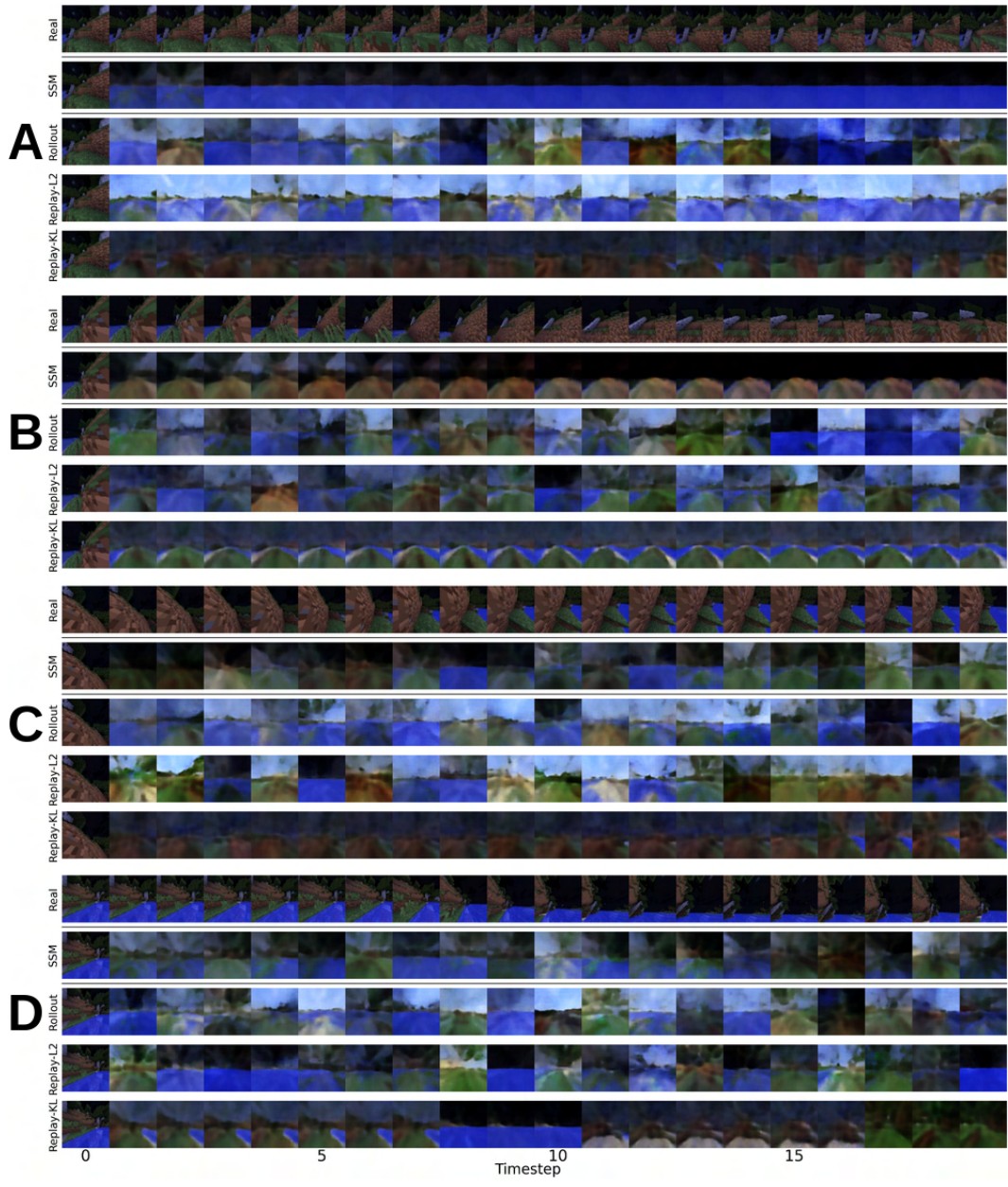

Figure 25: Examples from MineRL "Navigate-v0" task. (A) Long-horizon success; (B) long-horizon failure; (C) One-step success; (D) One-step failure.

package requires two `Distribution` objects to compute the KL distance between them *exactly*, hence making the computation quite expensive. However, if the type of distribution is set, efficiency can be improved by computing a closed-form solution and leveraging PyTorch parallel computation. Additionally, we remark that the test has been done in CPU, since all models could not fit in the VRAM of our GPU. However, when using only a single method, the computation can be parallelized in GPU to vastly improve speed.

Finally, we point out how encoding almost half a million transitions represents quite an extreme case needed only in open-ended tasks: in SuperTuxKart, for example, each dataset is composed of roughly 10k transitions; similarly, in Atari we store around 20k transitions per task. Therefore, we expect the time gap between methods to be quite limited in practice.

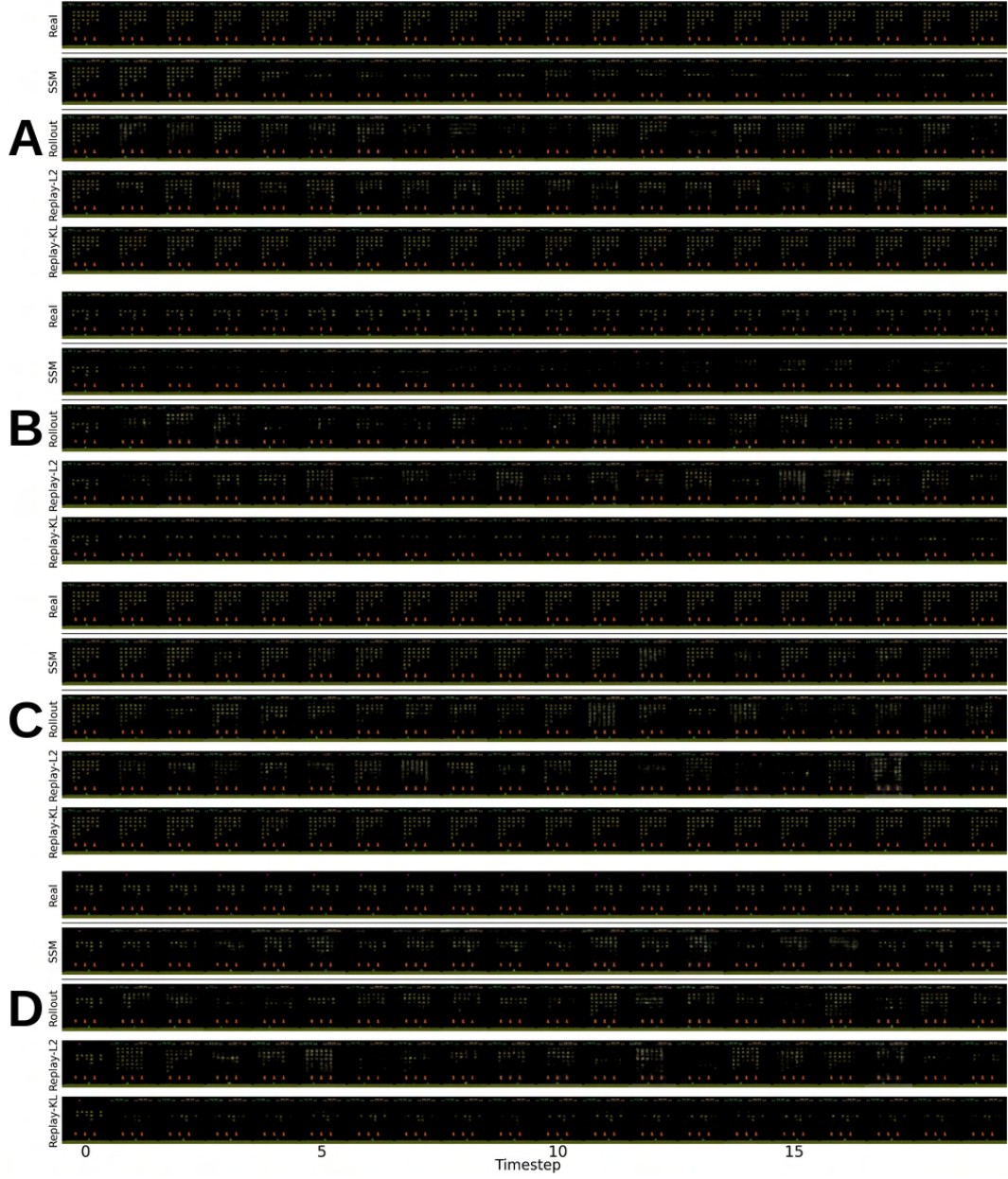

Figure 26: Examples from Atari "Space Invaders" games. (A) Long-horizon success; (B) long-horizon failure; (C) One-step success; (D) One-step failure.

To account for this expected practical scenario, we computed the inference time (dynamics evolution & action selection) for each model, by using a typical number of encoded trajectories for tasks in SuperTuxKart. As the inference time only depends on the number of encoded transitions, the results hold for any task using a comparable number of encodings of similar size. We show the results in Table 3.

In the "best-case scenario" of a small scoped task with a reasonable amount of encoded trajectories, PlaNet inference time is significantly higher than our methods. However, one might argue that in this case, performance could justify the gap if, for example, our approaches were to be comparable to a random policy. We address this point in Appendix **??** using data from the same experiment used to collect the numbers in Table 3.

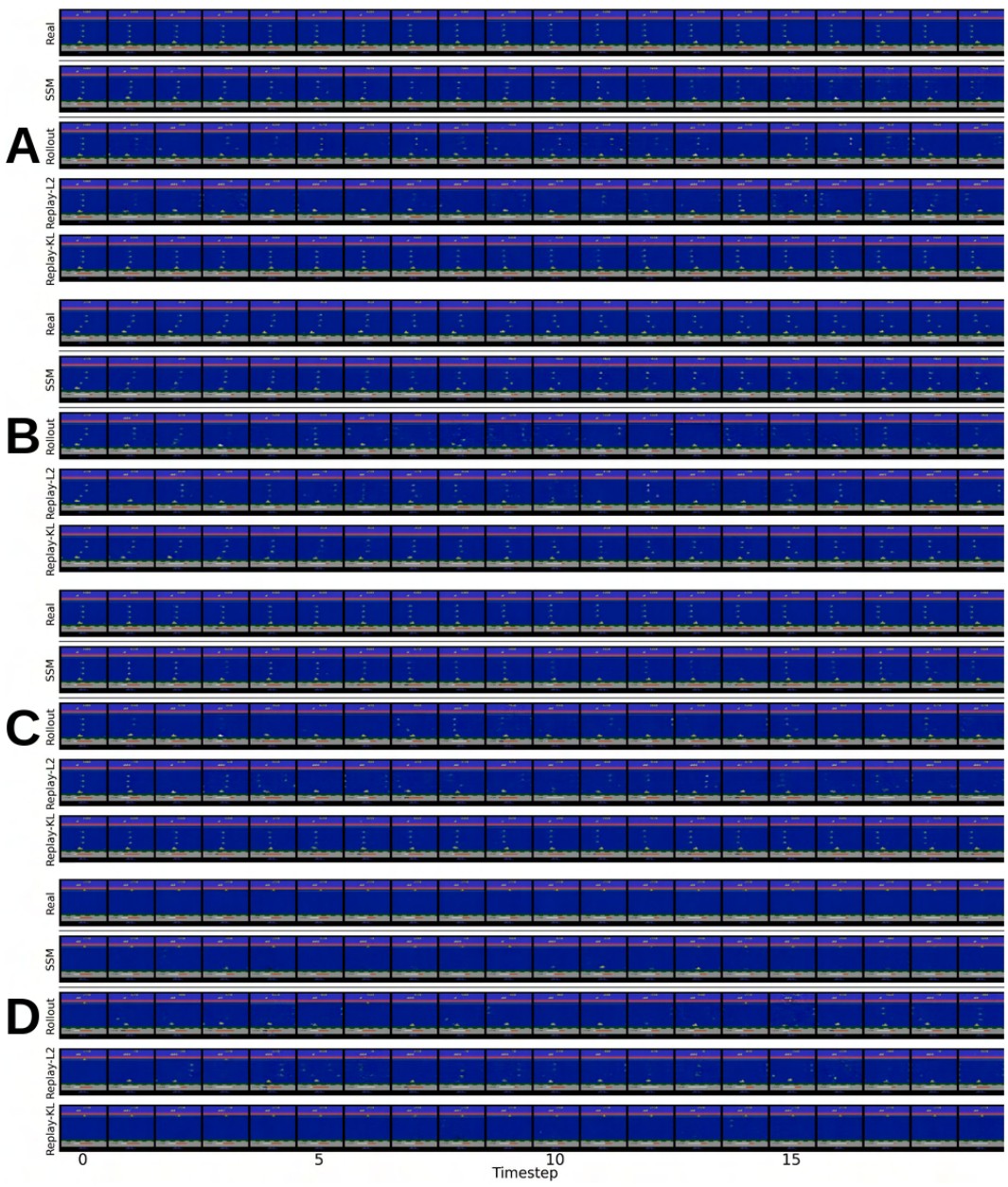

Figure 27: Examples from Atari "Seaquest" game. (A) Long-horizon success; (B) long-horizon failure; (C) One-step success; (D) One-step failure.

## C.2 Visual results

In Figure 29 we report an example of reconstruction for 10, 20, 40 and 80 encoded trajectories. The image suggests that encoding more images slightly improves visual resemblance in our method, mostly thanks to the more complete coverage of the state space. However, reconstruction is not always perfect. Interestingly, despite improving its latent prediction skills as reported in Section 5, the baseline cannot produce convincing decoded sequences from its predictions.

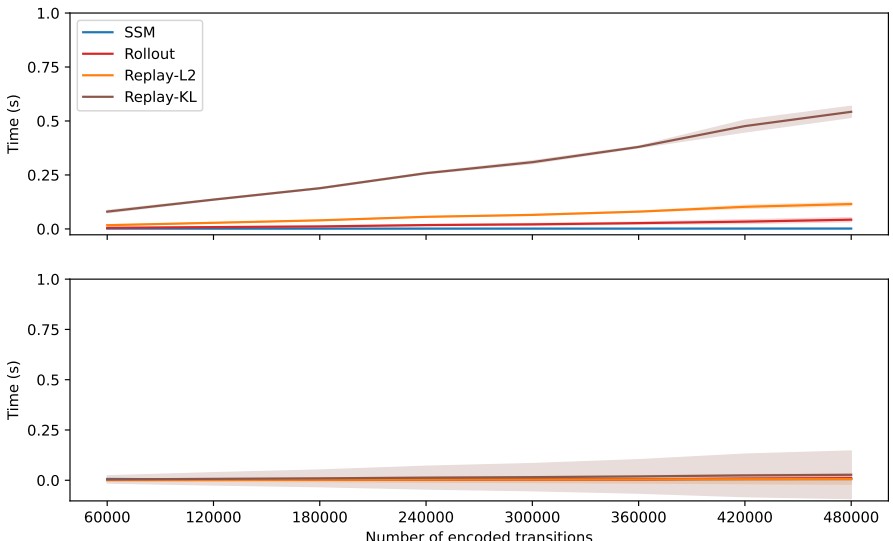

Figure 28: Average time needed for one-step (top) and long-horizon (bottom) latent dynamics prediction.

Table 3: Average inference time for each model in SuperTuxKart tasks. Results are consistent in all other environments, when using a similar number of encoded trajectories.

| Method | Inference time ($\mu \pm \sigma$) (s) |
|---|---|
| PlaNet | $0.04448 \pm 0.00392$ |
| Rollout | $0.00248 \pm 0.00021$ |
| L2 | $0.00084 \pm 0.00007$ |
| KL | $0.00188 \pm 0.00044$ |

## D    Action conditioning - visual examples

In Figures 30, 31, 32, 33 and 34 we report some visual examples of the effects of action conditioning for each track of SuperTuxKart. In each Figure, the first row shows the real sequence, while the three pairs of sequences show (top to bottom) Rollout, Replay-L2 and Replay-KL with (first row of a pair) and without (second row of a pair) action conditioning.

As reported in main text, conditioning on the action generally makes the prediction task harder, especially for Rollout and Replay-L2, as both models need to sample a batch on transitions to estimate the distribution. On the contrary, Replay-KL seems generally unaffected. We hypothesize that this is the result of matching the distribution rather than estimating it.

Moreover, in Figure 35 we report an example of retrieval failure for Replay-KL, as discussed in Section 5. It is notable how retrieving a significantly different first latent leads to a completely different predicted sequence. Fortunately, such occurrences are very rare, as discussed in Section 5.

## E    Algorithm & implementation details

In this Appendix, we report the architectures we have used for our study, along with some details and hyperparameters, to facilitate reproducibility of our results. A working implementation of our code is provided at `https://github.com/fmalato/zero_shot_world_models`.

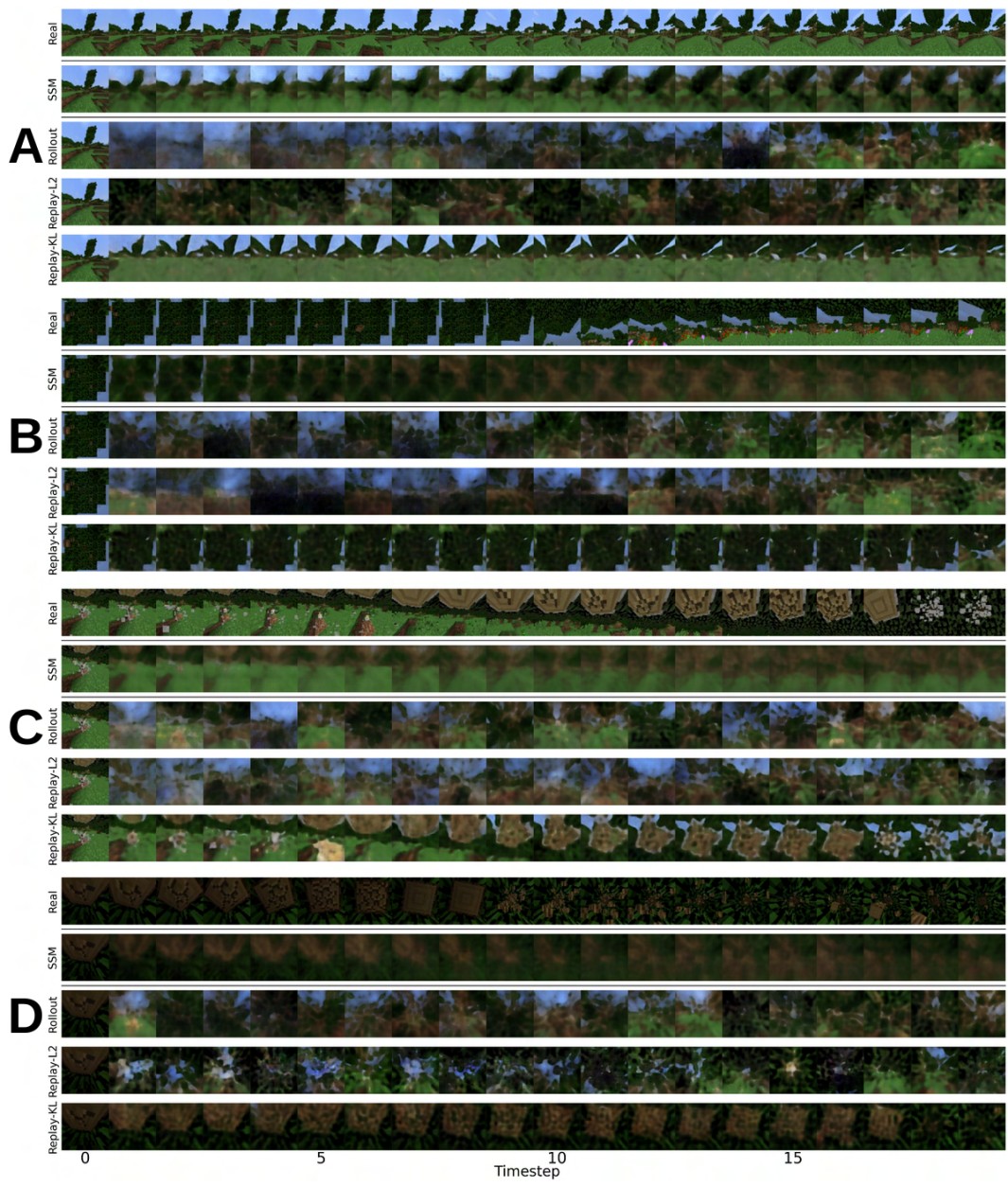

Figure 29: Visual examples of reconstructions from MineRL "Treechop-v0" using (A) ten, (B) twenty, (C) forty, and (D) eighty trajectories.

### E.1 VAE

Table 4 shows the hyperparameters used to train the VAEs. We train the models using the $\beta$-VAE variant, which includes a warm-up procedure for the KL divergence term. Specifically, the KL divergence part of the loss is masked in the first $10\%$ of the epochs, then it is gradually increased during the next $80\%$ of training time. The architecture of the encoder and decoder follows [8].

For MineRL, we have increased the latent size due to the higher amount of relevant information of an image. Additionally, we have decreased the number of epochs to $50$, as empirically the model showed signs of convergence around that time. Finally, we have increased the learning rate from $5 \times 10^{-5}$ to $3 \times 10^{-4}$, as it empirically led to the best results.

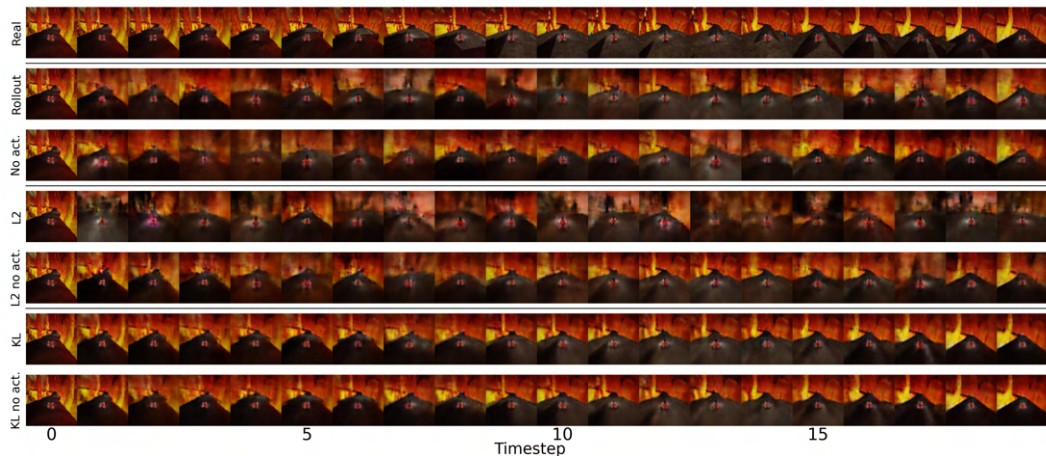

Figure 30: Effects of action conditioning on dynamics prediction in "fortmagma" track. First row reports the real sequence; after that, each pair of rows reports our three methods, respectively with and without action conditioning.

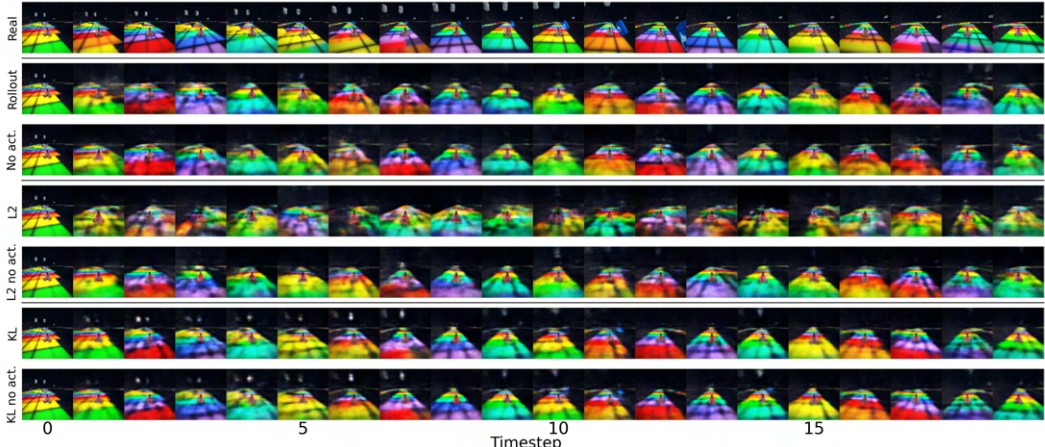

Figure 31: Effects of action conditioning on dynamics prediction in "snes_rainbowroad" track. First row reports the real sequence; after that, each pair of rows reports our three methods, respectively with and without action conditioning.

## E.2 SSM

We report the hyperparameters used to train the SSM model in Table 5. The training procedure follows [8]. From the original paper, we changed the values of the hyperparameters to obtain the best results, according to our empirical tests. Our PyTorch implementation uses `https://github.com/abhayraw1/planet-torch` as reference, even though the source code of the repo has been used only as a guide. We re-implemented the model as shown in the source code linked to this study.

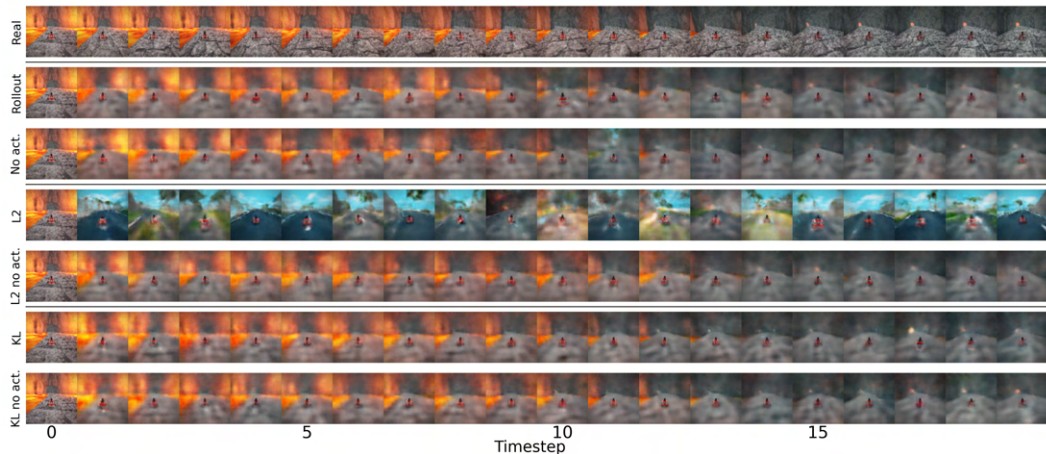

Figure 32: Effects of action conditioning on dynamics prediction in "volcano_island" track. First row reports the real sequence; after that, each pair of rows reports our three methods, respectively with and without action conditioning.

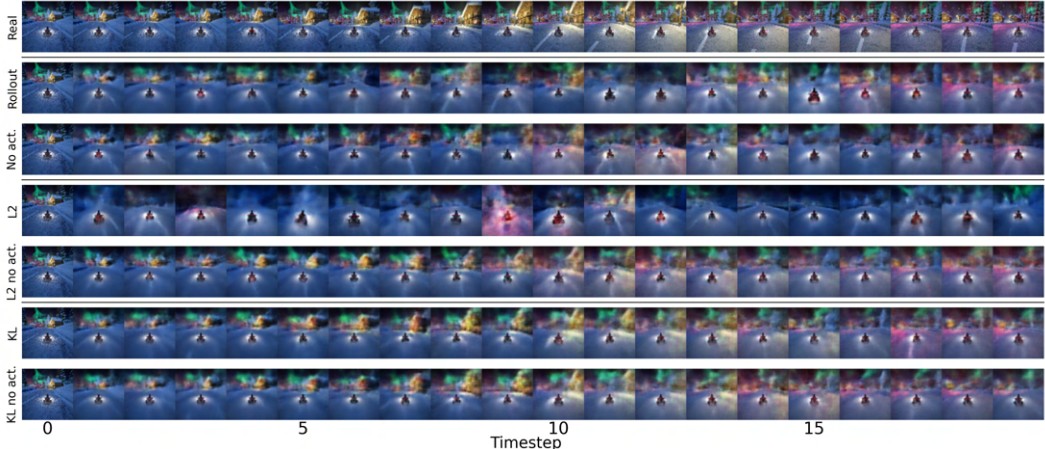

Figure 33: Effects of action conditioning on dynamics prediction in "snowmountain" track. First row reports the real sequence; after that, each pair of rows reports our three methods, respectively with and without action conditioning.

Table 4: Hyperparameters for VAE models used in this study.

| Name | SuperTuxKart | MineRL | Atari |
|---|---|---|---|
| image size | 64x64x3 | 64x64x3 | 64x64x3 |
| latent size | 128 | 512 | 128 |
| learning rate | $5 \times 10^{-5}$ | $3 \times 10^{-4}$ | $5 \times 10^{-5}$ |
| epochs | 250 | 50 | 250 |
| batch size | 128 | 128 | 128 |
| beta | $0.0 \to 5 \times 10^{-8}$ | $0.0 \to 5 \times 10^{-8}$ | $0.0 \to 5 \times 10^{-8}$ |
| beta interval (epochs) | $25 \to 225$ | $5 \to 45$ | $25 \to 225$ |

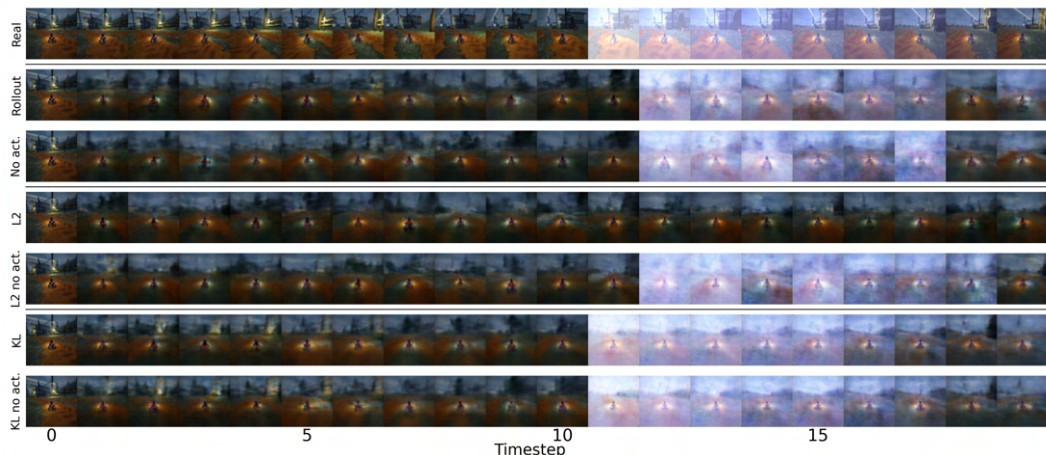

Figure 34: Effects of action conditioning on dynamics prediction in "lighthouse" track. First row reports the real sequence; after that, each pair of rows reports our three methods, respectively with and without action conditioning.

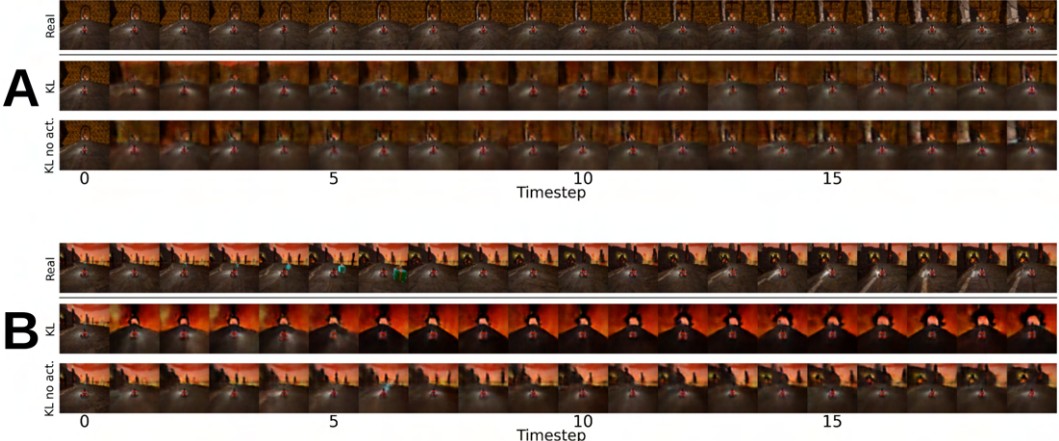

Figure 35: A comparison between (A) a successful retrieval and (B) a failed search by Replay-KL. Both sequences are extracted from the "fortmagma" track.

Table 5: Hyperparameters used to train the PlaNet baseline.

| Name | SuperTuxKart | MineRL | Atari |
|---|---|---|---|
| latent size | 128 | 512 | 128 |
| hidden size | 256 | 256 | 256 |
| learning rate | $1 \times 10^{-3}$ | $1 \times 10^{-3}$ | $1 \times 10^{-3}$ |
| epochs | 250 | 250 | 250 |
| batch size | 64 | 64 | 64 |
| beta | 0.1 | 0.1 | 0.1 |

