# OpenReview forum: "Zero-shot World Models via Search in Memory"
_NeurIPS.cc/2025/Conference — NeurIPS 2025 poster_

### Official Review · Reviewer_AMmz · 2025-06-21

**Clarity:** 2
**Significance:** 2
**Originality:** 2
**Rating:** 4
**Confidence:** 5

**Summary:**

This paper introduces a zero-shot world model that leverages similarity search and stochastic representations (via VAEs) to predict environment dynamics without explicit training. The approach encodes trajectories into a latent space, retrieves similar transitions, and probabilistically estimates future states. It is compared to PlaNet, a well-established model-based RL method, showing competitive performance in next-step and long-horizon (e.g., 20-step) predictions across diverse environments (SuperTuxKart, Minecraft, Atari). Key contributions include: (1) demonstrating the feasibility of memory-based world models, (2) benchmarking against learning-based models, and (3) identifying task-specific applicability and limitations.

**Questions:**

1. **Theoretical Justification**: What guarantees exist that retrieved local transitions approximate global dynamics? Is the Markov assumption empirically validated?
2. **Generalization Limits**: Why does performance degrade in Minecraft? Are trajectory coverage statistics (e.g., diversity metrics) correlated with errors?
3. **Action-Conditioning Dependency**: How would the model handle unknown future actions? Is there a mechanism to jointly optimize actions and dynamics?
4. **Efficiency Claims**: How does retrieval time scale with the trajectory count? A direct comparison with PlaNet’s training/inference costs is needed to validate "lightweight" claims.

**Ethical Concerns:**

["NO or VERY MINOR ethics concerns only"]

**Final Justification:**

Thank you to the authors for providing a thorough and constructive response to my initial review comments. After careful consideration of the additional empirical data and analysis you provided, I have decided to adjust my recommendation from "Reject" to "Borderline Reject."

While the new empirical analysis is valuable, there remains a lack of deeper theoretical explanation for why similarity search effectively approximates dynamics. For example, there is no analysis of how errors accumulate with prediction horizon length or under what conditions this method would fail.

While comparison with PlaNet is reasonable, the lack of comparison with more recent world model approaches (such as DreamerV3) limits the generalizability of the results.

**Limitations:**

Authors acknowledge data dependency, hallucinations, and variance in retrieval. However, they could quantify trajectory diversity requirements and propose mitigation strategies (e.g., causal priors or external knowledge integration).

**Paper Formatting Concerns:**

No issues

**Quality:**

2

**Strengths And Weaknesses:**

**Strengths** :

1. **Novelty**: Proposes a training-free framework for world modeling, diverging from traditional data-hungry approaches. This offers a lightweight alternative for resource-constrained scenarios.
2. **Empirical Validation**: Demonstrates competitive performance with PlaNet in long-horizon predictions (e.g., 20 steps) on visually complex environments, such as SuperTuxKart and Atari.
3. **Simplicity**: The method is computationally lightweight (deploying in seconds) and works with small trajectory datasets (e.g., five trajectories).

**Weaknesses** :

1. **Theoretical Gaps**: Lacks rigorous analysis of why similarity search approximates actual dynamics. The assumption of Markovian transitions is unproven, and error propagation mechanisms are unclear.
2. **Data Dependency**: Performance heavily relies on the quality and coverage of the trajectory. Results on Minecraft show significant degradation, indicating poor generalization in open-ended environments.
3. **Action Selection Limitation**: Assumes future actions are known, decoupling dynamics prediction from policy optimization. This restricts applicability to planning tasks where actions are externally provided.
4. **Reproducibility Concerns**: Implementation details (e.g., retrieval efficiency, hyperparameters) are sparse, and code and data availability are ambiguously stated.

---

> ### Author Rebuttal · Authors · 2025-07-30
>
> We thank the Reviewer for their insightful analysis of our manuscript and for the interesting questions raised. We provide the answers below, using an itemized list that follows the same order. At present time in the rebuttal process, it is not possible for us to share an updated version of the manuscript. However, changes to the manuscript following from the comments are marked for reference of the Reviewer. Please find our answers below.
>
> - Despite not having theoretical guarantees on the fact that retrieved local transitions approximate global dynamics, some reasonable assumptions used in our work could justify this fact. Firstly, the retrieval process implies that the error between an observation query $o$ and the corresponding retrieved observations $o'$ is minimized; assuming that such error is small enough, we could conclude that local transitions observed by a model approximate local transitions stored in the trajectories dataset $\mathcal{T}$. In second instance, considering that the encoded trajectories are collected from human gameplay, they will by definition implicitly follow the global dynamics of the environment; this is a main difference between our model and Dreamer-like architectures such as PlaNet, where the global dynamics are explicitly tracked by the hidden state of the RSSM. Finally, since the task of predicting dynamics with zero-shot world models can be reduced to compounding local transitions (that follow the Markov property) to estimate a generative distribution in a continuous latent space, we can reasonably assume that the retrieval process in its entirety follows the Markov property. Regarding the empirical validation of the property, we have not validated it.
>
> - We thank the Reviewer for highlighting this important point of discussion. To address it, we have computed the latent space binning coverage ratio using our test trajectories, and compared it to the KL error obtained by each model. Additionally, we computed the Pearson correlation coefficient for each model. As correctly stated by the Reviewer, after comparing the coverage statistics for SuperTuxKart and Minecraft, we found that lower coverage corresponds to higher mean error in transition retrieval. We report our results in Tables 1 & 2 (below), reporting respectively the dynamics prediction error (KL divergence) compared to the state space coverage ratio for each environment and model, and the Pearson correlation coefficient for the two quantities, for each model. Additionally, we have added the same results in Appendix A of our manuscript in visual form. Notably, all tested models show a strong correlation between data diversity and generalization capabilities. However, the correlation is particularly strong in PlaNet, accordingly with the well-known fact that neural networks greatly benefit from a more diverse dataset. Overall, our proposed models are slightly less dependent on data diversity.
>
> - We thank the Reviewer for raising this valid point. To answer this point we have performed an additional experiment, which we have also added to Appendix F of our manuscript. Please find the results in the Table 3 below. To account for unknown actions, we have compared the planning capabilities of each model. In particular, we let each model observe one frame every 20 steps and plan a sequence of actions up to the planning horizon; the actions are selected only based on the predicted dynamics. The results show that all our models yield comparable or superior performance to the PlaNet baseline, while requiring drastically less time to be applied (~15 seconds as opposed to ~5 hours for each environment, using same hardware). Furthermore, regarding the joint optimization of action and dynamics, we highlight that the only optimization procedure in our method consists of an independent training for the VAE encoder, which we operate following [1]. Aside from this, _none of our proposed methods requires any optimization_. The relevant processing needed to prepare our model for predicting dynamics is discussed in **Section 3** of our manuscript. Similarly, action selection and planning is discussed in **Section 3 (lines 117 - 120)**.
>
> - We thank the Reviewer for addressing this concern. We report the retrieval time for each model in Appendix C of our manuscript, where we test the feature under extreme conditions (Minecraft, 50k -> 500k encoded transitions). However, as addressed in Section 6 of our manuscript, we believet that our approach would be best suited for tasks with a more limited scope. As such, we conduct additional tests on the inference time of each model, that is, the time required to map an observation into an action. We have added these results in Appendix C, and we show them in Table 4 below. Finally, we agree with the Reviewer that claiming our method to be “lightweight” should be limited to such smaller scoped tasks, and have therefore updated our manuscript accordingly (**Section 6, lines 318 - 321**). For reference of the Reviewer, please find the added text below:
>
> **Added text:**
> > *Despite showing acceptable performance in open-ended tasks such as Minecraft and being comparable to the PlaNet baseline, our results suggest that the performance of zero-shot WMs depend on latent space coverage of the encoded trajectories, limiting their effectiveness in tasks with vast state spaces. However, they represent a valid, lightweight alternative to regular WMs in small-scoped tasks, such as SuperTuxKart and Atari.*
>
>
> ### Table 1: State space coverage ratio and dynamics prediction error (KL divergence) for the tested models on environments from SuperTuxKart and Minecraft.
>
> | Environment      | State Space Coverage Ratio |     PlaNet     | Rollout   | Replay-L2 | Replay-KL  |
> |------------------------|----------------------|-------------|-----------|-----------|------------|
> |TuxKart-lighthouse| 0.7675               |  571.3605   | 173.8899  | 192.4496  | 231.5324   |
> |TuxKart-fortmagma| 0.7300               |  519.0421   |  68.6900  |  86.9360  | 136.4911   |
> |TuxKart-rainbowroad| 0.8475               |  588.6913   | 146.5262  | 148.1211  | 521.2641   |
> |TuxKart-snowmountain| 0.6750               |  569.5750   |  67.0347  |  76.2962  | 133.6698   |
> |TuxKart-volcano_island| 0.5925               |  499.0704   |  85.8624  |  94.0208  | 217.8786   |
> |Minecraft-Treechop| 0.3950               | 2612.7190   | 394.5654  | 407.0560  | 1065.6871  |
> |Minecraft-Navigate| 0.6175               | 1156.5831   | 208.7204  | 196.5164  | 578.7786   |
>
> ### Table 2: Pearson correlation index of each model across the tested environments.
> | Model     | Pearson Correlation Coefficient |
> |-----------|---------------------|
> | SSM       | -0.8269             |
> | Rollout   | -0.6709             |
> | Replay-L2 | -0.6711             |
> | Replay-KL | -0.6366             |
>
> ### Table 3: Mean episodic return in SuperTuxKart environments. Each run is composed by 50 evaluation episodes.
>
> | Environment        | Model       | μ ± σ           | [CI 95%]           |
> |-------------------|-------------|------------------|--------------------|
> | **fortmagma**      | PlaNet      | 0.088 ± 0.113     | [0.056, 0.120]     |
> |                   | **Rollout**     | **5.810 ± 3.476**   | **[4.812, 6.808]** |
> |                   | Replay-L2   | 4.350 ± 1.451     | [3.934, 4.766]     |
> |                   | Replay-KL   | 4.774 ± 3.811     | [3.680, 5.868]     |
> | **lighthouse**     | PlaNet      | 3.078 ± 1.516     | [2.643, 3.513]     |
> |                   | Rollout     | 2.640 ± 2.595     | [1.895, 3.385]     |
> |                   | **Replay-L2**   | **5.114 ± 9.555**   | **[2.371, 7.857]** |
> |                   | Replay-KL   | 0.320 ± 0.756     | [0.103, 0.537]     |
> | **snes_rainbowroad** | PlaNet      | 3.160 ± 2.968     | [2.308, 4.012]     |
> |                   | Rollout     | 4.822 ± 1.119     | [4.501, 5.143]     |
> |                   | **Replay-L2**   | **6.336 ± 5.201**   | **[4.843, 7.829]** |
> |                   | Replay-KL   | 4.714 ± 0.652     | [4.527, 4.901]     |
> | **snowmountain**   | PlaNet      | 5.322 ± 3.788     | [4.235, 6.409]     |
> |                   | Rollout     | 5.556 ± 6.570     | [3.670, 7.442]     |
> |                   | Replay-L2   | 4.956 ± 6.504     | [3.089, 6.823]     |
> |                   | **Replay-KL**   | **15.664 ± 11.825** | **[12.269, 19.059]** |
> | **volcano_island** | PlaNet      | 0.928 ± 0.644     | [0.743, 1.113]     |
> |                   | Rollout     | 1.530 ± 0.585     | [1.362, 1.698]     |
> |                   | Replay-L2   | 3.348 ± 6.129     | [1.589, 5.108]     |
> |                   | **Replay-KL**   | **4.478 ± 7.965**   | **[2.191, 6.765]** |
>
> ### Table 4: Average inference time in SuperTuxKart environments.
>
> | Method  | Inference time ($\mu \pm \sigma$) (s)       |
> |---------|----------------------------------|
> | KL      | $0.00188 \pm 0.00044$           |
> | L2      | $0.00084 \pm 0.00007$           |
> | Rollout | $0.00248 \pm 0.00021$           |
> | PlaNet  | $0.04448 \pm 0.00392$           |
>
> ## References:
> [1] Diederik P Kingma, Max Welling, _Auto-Encoding Variational Bayes_, ICLR 2014.

---

> > ### Comment · Reviewer_AMmz · 2025-08-05
> >
> > Thank you to the authors for providing a thorough and constructive response to my initial review comments. After careful consideration of the additional empirical data and analysis you provided, I have decided to adjust my recommendation from "Reject" to "Borderline Reject."
> >
> > While the new empirical analysis is valuable, there remains a lack of deeper theoretical explanation for why similarity search effectively approximates dynamics. For example, there is no analysis of how errors accumulate with prediction horizon length or under what conditions this method would fail.
> >
> > While comparison with PlaNet is reasonable, the lack of comparison with more recent world model approaches (such as DreamerV3) limits the generalizability of the results.
> >
> > Add:

---

> > > ### Author Response · Authors · 2025-08-05
> > >
> > > We thank the Reviewer for their useful suggestions and for the constructive discussion. We regret to read that our explanation on error accumulation did not meet their expectation. In our view, Figures 2, 3 and 4 show how prediction error accumulates on variable time scales, for horizons up to 20 steps; additionally, we highlight that the error of a search does not affect the next retrieval, that is, error does not accumulate over different searches. For this reason we also believe that Figures 2-4 truthfully represent the reconstruction limits of our approach. However, we understand that longer horizons could have been considered for better evaluation of this aspect.
> > >
> > > For reference of the Reviewer, we would like to point out that the reason for not including more advanced models, such as DreamerV3, has been motivated in **lines 187 - 194** of our manuscript. Moreover, as stated in our previous comment, **lines 318 - 321** now include a clear explanation of where our method would fail, thanks to the suggestion of the Reviewers.

---

### Official Review · Reviewer_zx8U · 2025-06-22

**Clarity:** 3
**Significance:** 3
**Originality:** 3
**Rating:** 5
**Confidence:** 4

**Summary:**

In this paper the authors propose to use a world model zero shot via latent prediction with similarity search. They base their world model implementation on the PlaNet state space model [1],  from the family of Dreamer models [2].  The key innovation here is that rather than to learn a latent dynamics model as is done in [1], the authors here propose to design an algorithm that does this zero shot via similarity search [3] (a search method based upon a similarity metrics like L2, SSIM, KL etc.) and stochastic representations.

The stated main contributions of this work are:

 1. Demonstrate feasibility of  memory-based world models.
 2. Explore capabilities of said model and compare it to a training based baseline.
 3. Determined tasks for which this approach works well, and where it is infeasible.

The authors define their goal with the following problem statement:

"given a dataset D of state-action pairs (x_t, a_t) ∈ D, where xt is an RGB image representing the state of a system at timestep t ∈ [0, T], can we predict the transition dynamics P(x_t+1|x_t, a_t) of an environment without learning them?"

The authors observe that they can use similarity search over latent state space at t=t+1 to model z_t+1 effectively defining a latent state space transition model.  Thus given a set of trajectories each made up from successive transitions the set of latent transitions, Z, can be inferred over which search operations can be carried out to effect zero shot modeling of the state space model.  To test their method three model types are tested: *Rollout*, *Replay-L2*, *Replay-KL*.  *Rollout* stores independent latent trajectories which, when given a reference latent, can be used to estimate the next step latent distribution. *Replay* is simpler in that transitions independent of overall temporal dependence are retained and L2 and KL similarity variants are tested.

Baseline is the PlaNet state space model (SSM) trained on trajectories from their three model game domains (SuperTuxkart, Atari, Minecraft) which is compared to the Zero Shot model variants. The authors report KL, L1, and SSIM scores on the reconstructed trajectories along with the visual samples demonstrating top performance for *Rollout* & *Replay-KL*. The authors also investigate the effect of data volume and the effect action conditioning has on the models.

[1] Hafner, D., Lillicrap, T., Fischer, I., Villegas, R., Ha, D., Lee, H., and Davidson, J. (2018). Learning latent dynamics for planning from pixels. arXiv preprint arXiv:1811.04551.

[2]  Danijar Hafner, Timothy Lillicrap, Mohammad Norouzi, and Jimmy Ba. Mastering atari with discrete world models. arXiv preprint arXiv:2010.02193, 2020.

[3] https://en.wikipedia.org/wiki/Similarity_search

**Questions:**

Is the computational complexity of inference for the method discussed somewhere?

In section 5.2, the ablation study, it sounds like the rollout method performance is sensitive to the volume of data available while Replay-KL is more robust. Is there a sense, for any of the experimental domains, what this sensitivity looks like?  Can we infer from figure 6? Would this overall make Replay-KL the preferred method?

Regarding the third point of your stated contributions in the intro: "Determined tasks for which this approach works well, and where it is infeasible.", I didn't catch whether this is explicitly stated somewhere?  Could you clarify?

**Ethical Concerns:**

["NO or VERY MINOR ethics concerns only"]

**Final Justification:**

The authors have addressed issues around writing and presentation clarity and also on computational complexity of the method effectively. They have shown that world models can be leveraged zero shot through search methods to model environment dynamics and that the approach can also help with action selection.  I am supportive overall of the method which utilises world models in a novel fashion to to good effect and I am therefore happy to see this work published.

**Limitations:**

Not included by the authors directly, they should address this. They could probably discuss some things such as the dependence of their world models on good representations and exhibited some similarities to the chosen zero-shot domain, presumably drastically different world dynamics could lead to serious hallucinations.

**Quality:**

2

**Strengths And Weaknesses:**

**Strengths**

[Quality & Clarity] The overall structure of the paper is well organized and makes clear the primary goal of the work, directly stated at the beginning of section 3, and the related and relevant past work, the methods, the experiments and results. The writing is also for the most part clear and concise, and the description of the models and experiments are for the most part reasonable descriptions although I think there could be some more descriptive artifacts (more on this below).

[Originality] The authors build on past work done on Dreamer state-space models in a novel way. Using World Models zero shot by utilising a memory and similarity search is the main novelty of the paper and a good direction of inquiry that the authors appear to make work.  It is usually very common to use pre-trained language models zero shot so it's great to see this approach taken with a state-space model which can be expensive or tricky to optimise through training.

[Significance] World Models are a growing area of focus that are likely to be increasingly important as agent based applications become more ubiquitous.

**Weaknesses**

[Clarity] In section three I think a figure detailing how the rollout and replay algorithms function in detail would be helpful.  It could be a single figure that could help tie together the concepts throughout the entire section, since these ideas really drive the rest of your work conceptually. It may even be worth including an algorithmic element, but in my opinion at least one or the other would make things a lot clearer.  Further, I think a bit more detail about how the buffer datasets are composed could be helpful, for instance, some concrete indication of where the samples come from (this might be more appropriate for the experiments section).

I think it would be helpful to include some comparison of the computational burden of each approach at inference time, that is, baseline / rollout / replay approaches. It of course would be fair to consider any training resources for the baseline method but I think such considerations may be useful for these types of methods.

In section 3.1 it is stated that *"Then, we extract the next state and estimate the next latent distribution as previously discussed. "*.  As this is one of the best performing methods it'd be helpful to be specific about what was "previously discussed".

A bit more description of the SSM baseline training setup would be helpful if possible - and potentially some architectural details, although I suppose this can be found in the PlaNet paper, but it may be worth at least including in the appendix.

There is a typo in the second sentence of the abstract: "Improve" should be ""improved".

---

> ### Author Rebuttal · Authors · 2025-07-30
>
> We thank the Reviewer for their valuable comments. We will address each question in a separate item for clear readability, in the order provided by the Reviewer themselves. At present time in the rebuttal process, it is not possible for us to share an updated version of the manuscript. However, changes to the manuscript following from the comments are marked for reference of the Reviewer. Please find our answers below.
>
> - Computational complexity is discussed in Appendix C. We highlight that in the Appendix we evaluate how retrieval time scales with the number of encoded transitions (and, therefore, trajectories). However, we point out that this represents an unrealistic use case for our method, as according to our conclusions (Section 6, lines 318 - 321 & **Revised text** in item 2 of this answer), our method is best suited for small-scoped tasks, which will typically require a fraction of those transitions. To this end, we ran an additional experiment addressing action selection performance (added to Appendix F, Table 5 and shown in Table 1 below) on the tracks of the SuperTuxKart benchmark, in which we also computed the inference time for each agent (Table 2, below). We believe this test to be a much more interesting and likely scenario for our method. These results have been added also in Appendix C of our manuscript (Table 2). Furthermore, as a reference for the Reviewer, in this scenario the "training time" for our proposals consist of the time needed to encode trajectories, which is ~15 seconds; conversely, training the PlaNet baseline on the same hardware to obtain the results in Table 1 took ~5 hours for each environment.
>
> - We appreciate the comment and agree that a visual example showing the differences in reconstructing the dynamics w.r.t. the number of encoded trajectories would improve our manuscript. We have added a visualization of this fact in Appendix C of our updated version of the manuscript. To recover it while maintaining the visualization readable, we set the model to what we believe should be its standard use, namely long-term prediction without action conditioning. However, as correctly pointed out by the Reviewer, according to Figure 6 "Replay-KL" shows the most robust performance in predicting the evolution of the task dynamics over time. Relative to this, we highlight that the dynamics prediction does not necessarily correlate with better performance in planning, as shown in Table 1 (below). By testing the planning capabilities of our proposed methods, we show that while their performance is either on par or superior to the PlaNet baseline, no specific method dominates over the other across all tested tasks.
>
> - We agree with the Reviewer that the third point of our contribution could have been discussed more in detail. To this end, we have expanded our discussion on it in **Section 6, lines 318 - 321**. For reference of the Reviewer, please find our updated text below:
>
>   **Original text:**
>   > *Our method introduces three main benefits over the baseline: first, it lower the prediction error on latent dynamics, and either matches or lowers the reconstruction error for decoded trajectories; second, it improves consistency and visual appeal in long-horizon prediction; third, requires seconds to be implemented. Conversely, our method may suffer from hallucinations and lack of generalization when paired with insufficient data, as shown in the Minecraft benchmark. Additionally, the results obtained with our models may significantly vary, depending on the strategy used for retrieval.*
>
>   **Revised text:**
>   > *Our methods outperform the baseline on latent prediction and reconstruction errors, while also being more immediate in implementation and application. Despite showing acceptable performance in open-ended tasks such as Minecraft and being comparable to the PlaNet baseline, our results suggest that the performance of zero-shot WMs depend on latent space coverage of the encoded trajectories, limiting their effectiveness in tasks with vast state spaces. However, they represent a valid, lightweight alternative to regular WMs in small-scoped tasks, such as SuperTuxKart and Atari.*
>
> In addition to the questions posed by the Reviewer, we would like to briefly address the points listed under the "Weaknesses" category. Also in this instance we will answer following the order proposed by the Reviewer.
>
> - A Figure highlighting the differences among our proposed models has not been included due to page limits. However, we agree with the Reviewer that it would greatly help in conveying our proposal. As such, we are working on a solution to include one either in main text, or in the Appendices, while abiding by the submission limits.
>
> - A discussion on computational burden at inference time has been included above.
>
> - We thank the Reviewer for pointing out this shortcoming, and agree that the sentence is cryptic w.r.t. the relevance of the method. The sentence implicitly refers to Section 3, implying that after retrieving the 1-most similar transition from each trajectory, we leverage the minibatch of next observations {$o_{t+1,i}$}$_{i=1}^{|\mathcal{T}|}$ to estimate the latent distribution of the next sample, namely the next predicted step. For reference of the Reviewer, please find the updated text below:
>
> **Original text:**
>   > *Then, we extract the next state and estimate the next latent distribution as previously discussed.*
>
>   **Revised text:**
>   > *Then, we form the minibatch of next observations extracted by the current retrieval process. Finally, we estimate the next latent distribution by consideting each point in the minibatch as a realization of it, as discussed in Section 3.*
>
> - Some details regarding the training setup, e.g. hyperparameters, are provided in Appendix C, Tables 3 & 4. All the relevant architectural details are provided in the referenced papers. However, we agree with the Reviewer that providing the most relevant details in our Appendix could benefit to our manuscript, and are working on including them in Appendix C.
>
> - Typo fixed, we thank the Reviewer for spotting it.
>
> ### Table 1: Mean episodic return in SuperTuxKart environments. Each run is composed by 50 evaluation episodes.
>
> | Environment        | Model       | μ ± σ           | [CI 95%]           |
> |-------------------|-------------|------------------|--------------------|
> | **fortmagma**      | PlaNet      | 0.088 ± 0.113     | [0.056, 0.120]     |
> |                   | **Rollout**     | **5.810 ± 3.476**   | **[4.812, 6.808]** |
> |                   | Replay-L2   | 4.350 ± 1.451     | [3.934, 4.766]     |
> |                   | Replay-KL   | 4.774 ± 3.811     | [3.680, 5.868]     |
> | **lighthouse**     | PlaNet      | 3.078 ± 1.516     | [2.643, 3.513]     |
> |                   | Rollout     | 2.640 ± 2.595     | [1.895, 3.385]     |
> |                   | **Replay-L2**   | **5.114 ± 9.555**   | **[2.371, 7.857]** |
> |                   | Replay-KL   | 0.320 ± 0.756     | [0.103, 0.537]     |
> | **snes_rainbowroad** | PlaNet      | 3.160 ± 2.968     | [2.308, 4.012]     |
> |                   | Rollout     | 4.822 ± 1.119     | [4.501, 5.143]     |
> |                   | **Replay-L2**   | **6.336 ± 5.201**   | **[4.843, 7.829]** |
> |                   | Replay-KL   | 4.714 ± 0.652     | [4.527, 4.901]     |
> | **snowmountain**   | PlaNet      | 5.322 ± 3.788     | [4.235, 6.409]     |
> |                   | Rollout     | 5.556 ± 6.570     | [3.670, 7.442]     |
> |                   | Replay-L2   | 4.956 ± 6.504     | [3.089, 6.823]     |
> |                   | **Replay-KL**   | **15.664 ± 11.825** | **[12.269, 19.059]** |
> | **volcano_island** | PlaNet      | 0.928 ± 0.644     | [0.743, 1.113]     |
> |                   | Rollout     | 1.530 ± 0.585     | [1.362, 1.698]     |
> |                   | Replay-L2   | 3.348 ± 6.129     | [1.589, 5.108]     |
> |                   | **Replay-KL**   | **4.478 ± 7.965**   | **[2.191, 6.765]** |
>
> ### Table 2: Average inference time in SuperTuxKart environments.
>
> | Method  | Inference time ($\mu \pm \sigma$) (s)       |
> |---------|----------------------------------|
> | KL      | $0.00188 \pm 0.00044$           |
> | L2      | $0.00084 \pm 0.00007$           |
> | Rollout | $0.00248 \pm 0.00021$           |
> | PlaNet  | $0.04448 \pm 0.00392$           |

---

> > ### Comment · Reviewer_zx8U · 2025-08-04
> > **Rebuttal Response**
> >
> > Thanks for taking some of these suggestions into consideration and even going to the extra effort to run and include some new empirical results.  I believe that by addressing some issues on the paper clarity and computational complexity the impact of the manuscript overall will be improved and that the applicability of the approach will be clearer.  I am supportive of the method which utilises world models in a novel fashion to to good effect and I am therefore happy to raise my score up to an 'accept' to reflect the proposed improvements.

---

> > > ### Author Response · Authors · 2025-08-05
> > >
> > > We thank the Reviewer for the time they spent into evaluating our manuscript and for their valuable input.

---

### Official Review · Reviewer_JymM · 2025-07-01

**Clarity:** 3
**Significance:** 1
**Originality:** 3
**Rating:** 5
**Confidence:** 4

**Summary:**

This paper introduces a method of approximating a world model without training a world model leveraging similarity search and stochastic representations. This method does similarity search in a latent memory buffer constructed using a pretrained variational autoencoder (VAE). The authors introduce three different memory structures and compare these against PlaNet, using reconstruction quality in latent and image space as their evaluation metric. The paper reports competitive or better results than PlaNet, evaluated on SuperTuxKart, Minecraft and Atari Seaquest and Space Invaders.

The aim of this paper is to introduce an alternative formulation of world models that does not require massive amounts of data and training.

**Questions:**

I don't have any specific questions. Overall I feel that in the current form, the work does not provide enough evidence to reach the self-stated goal of offering a feasible alternative to existing world models. The work needs additional evaluation that effectively stress-tests similarity search and shows the feasibility of this method in a decision making process.

**Ethical Concerns:**

["NO or VERY MINOR ethics concerns only"]

**Final Justification:**

Given the new results and other changes the authors have provided I think this is an interesting direction of work that has been well-motivated and effectively evaluated.

**Limitations:**

The authors don't note their assumption of image reconstruction as a sufficient proxy for world models.

**Quality:**

1

**Strengths And Weaknesses:**

**Strengths**
- The proposal of representing a world model using only memory search seems novel and appealing because of the huge requirement of resources of existing world models.
- The paper is well written and builds well to form an understandable explanation of the proposed method. The paper also has a detailed description of the experiments and results.
- The authors propose three different memory structures and evaluate all memory structures in their experiments.

**Weaknesses**
- the third contribution stated in the introduction is "third, we determine a range of tasks for which such models are applicable, and clearly state situations where they are unfeasible." However, the paper does not clearly establish where the method fails or systematic analysis of feasibility, instead only pointing out that their method struggles with Minecraft because of insufficient data but this is far from clearly establishing where the method fails.
-  The chosen benchmarks are favourable for similarity search and I feel they do not effectively stress test similarity search vs a learnt world model. For example, the authors design a memory buffer that can account for actions that result in huge visual differences in future states (the most obvious use with the proposed method) but do not evaluate on environments where this occurs.
- The entire evaluation of this work is based on the assumption that high-fidelity image reconstruction is a good proxy for a good world model. However, world models are used for task learning so a model that filtered out task irrelevant features would be a better world model then a perfect pixel-for-pixel reconstruction. I would argue that using only image reconstruction as the evaluation metric is not sufficient to determine if this method is a valid alternative to existing world models.
- Following from the previous point, the authors compare against PlaNet by comparing the quality of image reconstruction between their method and PlaNet ("benchmark specific aspects of the compared models, namely the ability to reconstruct in a seemingly infinite observation space, and the ability to focus on small details."). However, the PlaNet design is optimized for policy learning and control and so claiming competitve or superior results over PlaNet seems disingenuous without including policy-learning experiments.
- The authors point out that they rely on fixed actions and claim this is okay because they are only focused on the quality of dynamics prediction over time. However, the absence of exploring how their method handles autonomous decision making limits the paper's relevance. Without evaluating this method in a decision making process I don't think the authors can claim this work is a valid alternative to existing world models.

---

> ### Author Rebuttal · Authors · 2025-07-30
>
> We thank the Reviewer for their insightful analysis and provide discussion points to the “Weaknesses” listed by them, following the same order. At present time in the rebuttal process, it is not possible for us to share an updated version of the manuscript. However, changes to the manuscript following from the comments are marked for reference of the Reviewer. Please find our answers below.
>
> -  We recognize that the text was previously unclear about the limitations of our method. To address this, we updated **Section 6 (lines 317–321)** to better articulate both the strengths and shortcomings. Below is the revised text for the Reviewer's reference:
>
>   **Original text:**
>   > *Our method introduces three main benefits over the baseline: first, it lower the prediction error on latent dynamics, and either matches or lowers the reconstruction error for decoded trajectories; second, it improves consistency and visual appeal in long-horizon prediction; third, requires seconds to be implemented. Conversely, our method may suffer from hallucinations and lack of generalization when paired with insufficient data, as shown in the Minecraft benchmark. Additionally, the results obtained with our models may significantly vary, depending on the strategy used for retrieval.*
>
>   **Revised text:**
>   > *Our methods outperform the baseline on latent prediction and reconstruction errors, while also being more immediate in implementation and application. Despite showing acceptable performance in open-ended tasks such as Minecraft and being comparable to the PlaNet baseline, our results suggest that the performance of zero-shot WMs depend on latent space coverage of the encoded trajectories, limiting their effectiveness in tasks with vast state spaces. However, they represent a valid, lightweight alternative to regular WMs in small-scoped tasks, such as SuperTuxKart and Atari.*
>
>
> - We thank the Reviewer for raising this point. We would like to highlight that tasks from Minecraft are substantially unpredictable due to the highly diverse observations that an agent could encounter. These tasks have been selected precisely to test the limits of similarity search on vast state spaces and have been tested coherently with the other selected benchmarks. Examples of this additional difficulty can be seen by comparing the values in Figures 2, 3, and 4 from our manuscript: in general, error values in Figure 3 (related to Minecraft tasks) are significantly higher than in the other two benchmarks, suggesting that, as expected, all models struggle more in predicting dynamics from Minecraft. However, we would appreciate it if the Reviewer could provide some examples of environments deemed to be suitable for this point.
>
> - We thank the Reviewer for providing this comment. To answer, we would like to highlight that our methods are evaluated both in the _visual domain_ through L1 and SSIM reconstruction error and in _latent space_ through KL divergence *(lines 203 - 206)*, which account for the *dynamics prediction error*. Moreover, we specify that L1 and SSIM are necessarily affected by decoding errors and, as such, should be considered in conjunction with KL and the visual examples *(lines 207 - 210)*.
>
> - We agree that testing the action selection performance would be beneficial to our manuscript. Therefore, we have added an experiments that compares the models during planning. The test is designed as follows: on the first timestep of an episode, each model observes the environment. Then, based solely on this observation, models are asked to evolve the dynamics using their world model and plan 20 actions accordingly. The actions are then executed as predicted, giving models no chance to change their plan. Finally,  We report the results in Table 1 (below) as _mean_ $\pm$ _std_, and use a 95% confidence interval as additional metric. We add the same results in our manuscript in Appendix F (Table 5). We highlight that the test was run only on the SuperTuxKart environments due to time restrictions. The results show that action selected by our models are, on average, either comparable or better than the one chosen by the selected baseline. For reference of the Reviewer, these results were obtained by encoding 10 trajectories for each task (time taken: ~15 seconds) for our methods, as opposed to training a PlaNet baseline for 25 epochs (time taken: ~5 hours). All methods were tested using the same hardware. In Table 2 (below) we also report the average inference time for each model, as recorded during the same test.
>
> - We believe that the additional experiment highlithed in the previous point may be sufficient to cover this discussion point as well.
>
> ### Table 1: Mean episodic return in SuperTuxKart environments. Each run is composed by 50 evaluation episodes.
>
> | Environment        | Model       | μ ± σ           | [CI 95%]           |
> |-------------------|-------------|------------------|--------------------|
> | **fortmagma**      | PlaNet      | 0.088 ± 0.113     | [0.056, 0.120]     |
> |                   | **Rollout**     | **5.810 ± 3.476**   | **[4.812, 6.808]** |
> |                   | Replay-L2   | 4.350 ± 1.451     | [3.934, 4.766]     |
> |                   | Replay-KL   | 4.774 ± 3.811     | [3.680, 5.868]     |
> | **lighthouse**     | PlaNet      | 3.078 ± 1.516     | [2.643, 3.513]     |
> |                   | Rollout     | 2.640 ± 2.595     | [1.895, 3.385]     |
> |                   | **Replay-L2**   | **5.114 ± 9.555**   | **[2.371, 7.857]** |
> |                   | Replay-KL   | 0.320 ± 0.756     | [0.103, 0.537]     |
> | **snes_rainbowroad** | PlaNet      | 3.160 ± 2.968     | [2.308, 4.012]     |
> |                   | Rollout     | 4.822 ± 1.119     | [4.501, 5.143]     |
> |                   | **Replay-L2**   | **6.336 ± 5.201**   | **[4.843, 7.829]** |
> |                   | Replay-KL   | 4.714 ± 0.652     | [4.527, 4.901]     |
> | **snowmountain**   | PlaNet      | 5.322 ± 3.788     | [4.235, 6.409]     |
> |                   | Rollout     | 5.556 ± 6.570     | [3.670, 7.442]     |
> |                   | Replay-L2   | 4.956 ± 6.504     | [3.089, 6.823]     |
> |                   | **Replay-KL**   | **15.664 ± 11.825** | **[12.269, 19.059]** |
> | **volcano_island** | PlaNet      | 0.928 ± 0.644     | [0.743, 1.113]     |
> |                   | Rollout     | 1.530 ± 0.585     | [1.362, 1.698]     |
> |                   | Replay-L2   | 3.348 ± 6.129     | [1.589, 5.108]     |
> |                   | **Replay-KL**   | **4.478 ± 7.965**   | **[2.191, 6.765]** |
>
> ### Table 2: Average inference time in SuperTuxKart environments.
>
> | Method  | Inference time ($\mu \pm \sigma$) (s)       |
> |---------|----------------------------------|
> | KL      | $0.00188 \pm 0.00044$           |
> | L2      | $0.00084 \pm 0.00007$           |
> | Rollout | $0.00248 \pm 0.00021$           |
> | PlaNet  | $0.04448 \pm 0.00392$           |

---

> > ### Comment · Reviewer_JymM · 2025-07-31
> > **Response**
> >
> > Thank you for addressing the points and these additional experiments.
> >
> > While I can appreciate Minecraft is a challenging domain I don't think it fully tests how your method would cope under extreme changes in the environment due to one, or few, actions.
> > > However, we would appreciate it if the Reviewer could provide some examples of environments deemed to be suitable for this point.
> >
> > An example may be something like the Atari Montezuma's Revenge or Pitfall where the agent can transition between screens in a single action which drastically changes the appearance of the domain. I think an example of such a domain would greatly strengthen the results as this is a point that I would imagine a replay-buffer based world model would suffer.
> >
> > With that said, in light of the new results I will be changing my score. Previously, the experiments did not convince me that your method would be a suitable replacement for a world model. However, by including the action selection step I believe this work shows a promising new direction for world models and worth publishing.
> >
> > I also appreciate the change to the limitations section.

---

> > > ### Author Response · Authors · 2025-08-05
> > >
> > > We thank the Reviewer for their time and their valuable suggestions. We also cherish their advice on the environment choice and will keep it in consideration for the future.

---

### Official Review · Reviewer_4hop · 2025-07-03

**Clarity:** 3
**Significance:** 2
**Originality:** 3
**Rating:** 4
**Confidence:** 3

**Summary:**

This paper leverages similarity search and stochastic representations to approximate a world model without a training procedure. Comparison experiments with PlaNet showed comparable quality of latent reconstruction and reconstructed images. The model shows stronger performance in long-horizon prediction with respect to the baseline on a range of visually different environments. Besides, the authors mentioned a possible limitation of this method: "Conversely, our method may suffer from hallucinations and lack of generalization when paired with insufficient data, as shown in the Minecraft benchmark."

**Questions:**

Please see Strengths and Weaknesses above.

**Ethical Concerns:**

["NO or VERY MINOR ethics concerns only"]

**Final Justification:**

I have carefully read the author's rebuttal. The reviewer greatly appreciates the additional experiments and explanations regarding novelty provided by the author. I will raise my score.

**Limitations:**

yes

**Quality:**

2

**Strengths And Weaknesses:**

Strengths:
- The paper shows a new Zero-shot World Model without a training procedure. The reviewer is not very familiar with this field, but feels that this setting is still very meaningful.
- Comparison experiments with PlaNet showed comparable quality of latent reconstruction and reconstructed images.
- The paper is well-written and easy to follow.

Weaknesses:
- The images generated by the world model are very blurry and cannot be clearly seen by the human eye. I am not sure whether this generation quality can provide any useful information for the model, or how the action model can benefit from this world model.
- The experiments only show the results on generation metrics, KL divergence, L1 distance and SSIM. How the action model utilizes the world model has not been evaluated.
- (minor) I am quite curious about the comparison and discussion between this type of world model and the method of using diffusion to achieve video generation as a world model.

In all, if the author can thoroughly address my concern, I am very willing to raise my score.

---

> ### Author Rebuttal · Authors · 2025-07-30
>
> We thank the Reviewer for their valuable suggestions and gladly answer the interesting points they raised. We address each question in a separate item, following the order specified by the Reviewer. At present time in the rebuttal process, it is not possible for us to share an updated version of the manuscript. However, changes to the manuscript following from the comments are marked for reference of the Reviewer. Please find our answers below.
>
> - We agree that the images are not extremely clear and may appear blurry; however, we highlight that the images shown in the manuscript are intended for visual evaluation purposes only. Additionally, most of the images are VAE reconstructions of the predicted dynamics, which are notoriously affected by a small reconstruction error causing the blur. Regarding the informative content of the images, we agree with the Reviewer that greater resolution could help in preserving details and hence add meaningful information for a model. However, we point out that our experimental protocol follows common practices found in relevant previous literature, such as using 84x84 [1], or 64x64 [2, 3] images for learning to act & plan with partial pixel observations.
>
> - We agree with the Reviewer that comparing the models for action selection and planning would constitute a valuable addition to our manuscript. Therefore, we have designed an additional experiment to test the action models. The experiment is designed as follows: each model is tested on the realized returns over 50 evaluation episodes. Each model is allowed to observe the first state of an episode. From this single observation, agents are asked to evolve the dynamics according to their world model and plan the next 20 actions. Then, the planned actions are executed with no room for corrections. Finally, the model is allowed to observe a new state and plan the next sequence of actions. The process is repeated until the end of the episode. We report the results in Table 1 (below) using both _mean_ $\pm$ _std_ and [95% confidence interval]. Additionally, we have reported the same results in our manuscript in Appendix F, Table 5. We highlight that the test was run only on the SuperTuxKart environments due to time restrictions. The results show that zero-shot world models consistently outperform the PlaNet baseline, even though no specific approach shows consistently superior performance. These new results support our conclusion that zero-shot world models can be a valid alternative to learning-based ones.
>
> - We thank the Reviewer for providing such an interesting point of discussion. Despite the vast gap between the two ideas, we believe that a diffusion model would indeed achieve superior performance to our methods. However, it is worth noting that, with respect to our proposal _that requires no training at all_, a diffusion model would need a significant amount of data and time to learn the dynamics of each environment. With our study, we aim at showing that "lightweight" alternatives to computationally heavy models might exist. Therefore, despite recognizing the validity of the proposal advanced by the Reviewer, we believe that a comparison between the two approaches would not be fair due to the vast gap in computational resources needed.
>
> ### Table 1: Mean episodic return in SuperTuxKart environments. Each run is composed by 50 evaluation episodes.
>
> | Environment        | Model       | μ ± σ           | [CI 95%]           |
> |-------------------|-------------|------------------|--------------------|
> | **fortmagma**      | PlaNet      | 0.088 ± 0.113     | [0.056, 0.120]     |
> |                   | **Rollout**     | **5.810 ± 3.476**   | **[4.812, 6.808]** |
> |                   | Replay-L2   | 4.350 ± 1.451     | [3.934, 4.766]     |
> |                   | Replay-KL   | 4.774 ± 3.811     | [3.680, 5.868]     |
> | **lighthouse**     | PlaNet      | 3.078 ± 1.516     | [2.643, 3.513]     |
> |                   | Rollout     | 2.640 ± 2.595     | [1.895, 3.385]     |
> |                   | **Replay-L2**   | **5.114 ± 9.555**   | **[2.371, 7.857]** |
> |                   | Replay-KL   | 0.320 ± 0.756     | [0.103, 0.537]     |
> | **snes_rainbowroad** | PlaNet      | 3.160 ± 2.968     | [2.308, 4.012]     |
> |                   | Rollout     | 4.822 ± 1.119     | [4.501, 5.143]     |
> |                   | **Replay-L2**   | **6.336 ± 5.201**   | **[4.843, 7.829]** |
> |                   | Replay-KL   | 4.714 ± 0.652     | [4.527, 4.901]     |
> | **snowmountain**   | PlaNet      | 5.322 ± 3.788     | [4.235, 6.409]     |
> |                   | Rollout     | 5.556 ± 6.570     | [3.670, 7.442]     |
> |                   | Replay-L2   | 4.956 ± 6.504     | [3.089, 6.823]     |
> |                   | **Replay-KL**   | **15.664 ± 11.825** | **[12.269, 19.059]** |
> | **volcano_island** | PlaNet      | 0.928 ± 0.644     | [0.743, 1.113]     |
> |                   | Rollout     | 1.530 ± 0.585     | [1.362, 1.698]     |
> |                   | Replay-L2   | 3.348 ± 6.129     | [1.589, 5.108]     |
> |                   | **Replay-KL**   | **4.478 ± 7.965**   | **[2.191, 6.765]** |
>
> ## References:
> [1] Volodymyr Mnih, Koray Kavukcuoglu, David Silver, Alex Graves, Ioannis Antonoglou, Daan Wierstra, Martin A. Riedmiller, _Playing Atari with Deep Reinforcement Learning._ CoRR abs/1312.5602 (2013)
>
> [2] Danijar Hafner, Timothy P. Lillicrap, Ian Fischer, Ruben Villegas, David Ha, Honglak Lee, James Davidson, _Learning Latent Dynamics for Planning from Pixels._ ICML 2019: 2555-2565
>
> [3] Danijar Hafner, Timothy P. Lillicrap, Mohammad Norouzi, Jimmy Ba, _Mastering Atari with Discrete World Models._ ICLR 2021

---

> > ### Comment · Reviewer_4hop · 2025-08-04
> >
> > I have carefully read the author's rebuttal. The reviewer greatly appreciates the additional experiments and explanations regarding novelty provided by the author. I will raise my score.

---

> > > ### Author Response · Authors · 2025-08-05
> > >
> > > We thank the Reviewer once again for their valuable input and for their time.

---

### Note · Authors · 2025-08-14

As a final remark, we would like to thank the reviewers for their interesting and valuable suggestions. We are thrilled to read that Reviewers consider our idea novel, relevant, and clearly explained. We highlight three particularly important improvements following from the comments:
- Reviewers `JymM` and `zx8U` suggested to clarify the limitations and failure cases of our approach. Following their advice, the camera-ready version of our manuscript will reflect these changes in **lines 318 - 321**.
- Reviewers `4hop`, `JymM` and `AMmz` highlighted that the quality of our work could be improved by assessing the planning and action selection performance of our approach w.r.t. the baseline. As reported during the discussion, this experiment has been run and the results will appear in **Appendix F** of the final version of the manuscript.
- Reviewers `zx8U` and `AMmz` raised concerns on the inference time of our model due to the retrieval process. We addressed this comment in the discussion, by providing additional results on inference time during action selection. This new result will be added to the camera-ready version of the manuscript, expanding the study in **Appendix C.1**.

Also, a number of additional suggestions have been considered and added to the manuscript, as reported in our comments to the Reviewers. These changes will be visible in the final version of the paper. In thanking once again the Reviewers for their time, we hope that our rebuttals have been satisfactory.

---

### Decision · Program_Chairs · 2025-09-17

**Decision:**

Accept (poster)

**Comment:**

This paper proposes a zero-shot world model that leverages similarity search over a latent memory buffer and stochastic representations via a pretrained VAE to predict environment dynamics without explicit training. The approach stores trajectories and retrieves similar transitions to estimate future states, enabling long-horizon predictions in diverse environments such as SuperTuxKart, Minecraft, and Atari.

The paper is practical and well-written, offering a lightweight alternative to traditional data-hungry world models. Strengths include its simplicity, empirical validation across multiple domains, and a clear description of memory structures and experiments. Weaknesses include limited theoretical analysis, reliance on known future actions, sensitivity to dataset quality, and incomplete evaluation of downstream control tasks. Reviewers also suggested clarifying algorithmic steps, memory buffer composition, and minor implementation details.

During rebuttal, the authors addressed key concerns by providing additional planning experiments, clarifying limitations, and expanding methodological discussions. These updates demonstrate that the method is usable as a world model and highlight future research directions. Considering the strong response to reviewer concerns, I recommend accept (Consensus Reached), with the requirement that the final version includes the promised clarifications, supplementary experiments, and minor corrections.